# ReIn: Conversational Error Recovery with Reasoning Inception ✦

**Takyoung Kim**[1*]  **Jinseok Nam**[2]  **Chandrayee Basu**[2]  **Xing Fan**[2]  **Chengyuan Ma**[2]
**Heng Ji**[1]  **Gokhan Tur**[1]  **Dilek Hakkani-Tür**[1]
[1]University of Illinois Urbana-Champaign    [2]Amazon
tk30@illinois.edu

## ABSTRACT

Conversational agents powered by Large Language Models (LLMs) with tool integration achieve strong performance on fixed task-oriented dialogue datasets but remain vulnerable to unanticipated, user-induced errors. Rather than focusing on error prevention, this work focuses on error recovery, which necessitates the accurate diagnosis of erroneous dialogue contexts and execution of proper recovery plans. Under realistic constraints precluding model fine-tuning or system prompt modification due to significant cost and time requirements, we explore whether agents can recover from contextually flawed interactions and how their behavior can be adapted without altering model parameters and prompts. To this end, we propose **Reasoning Inception (ReIn)**[1], a test-time intervention method that *plants* an initial reasoning into the agent's decision-making process. Specifically, an external inception module identifies predefined errors within the dialogue context and generates recovery plans, which are subsequently integrated into the agent's internal reasoning process to guide corrective actions, without modifying its parameters or system prompts. We evaluate ReIn by systematically simulating conversational failure scenarios that directly hinder successful completion of user goals: user's ambiguous and unsupported requests. Across diverse combinations of agent models and inception modules, ReIn substantially improves task success and generalizes to unseen error types. Moreover, it consistently outperforms explicit prompt-modification approaches, underscoring its utility as an efficient, on-the-fly method. In-depth analysis of its operational mechanism, particularly in relation to instruction hierarchy, indicates that jointly defining recovery tools with ReIn can serve as a safe and effective strategy for improving the resilience of conversational agents without modifying the backbone models or system prompts.

## 1 INTRODUCTION

Large Language Model (LLM)-based conversational agents can directly engage users and invoke relevant tools, achieving strong performance across many tasks (Bubeck et al., 2023; Yehudai et al., 2025; Guan et al., 2025). Yet they still remain susceptible to limitations such as hallucinations (Li et al., 2023), misinterpretation of long context (Laban et al., 2025), and failure in maintaining consistency (Jang & Lukasiewicz, 2023). These errors can arise unpredictably during multi-turn interactions, highlighting the gap between general capabilities and real-world deployment.

Beyond agent-side limitations, an underappreciated source of failure is the *user*. In spoken conversational systems, users often struggle to clearly express or even identify their own intents. A useful way to think about this is through a tutor-student analogy: much like a co-pilot guiding a novice learner, the agent must detect potential errors in real time and provide corrective guidance to keep the interaction on track. Without such support, user's unexpected requests can easily derail the conversation, leaving user goals unsatisfied. Prior agent-side mitigations, such as clarification (Min et al., 2020; Keyvan & Huang, 2022; Li et al., 2024a; Zhang & Choi, 2025) and fallback mechanisms (Shrivastava et al., 2021; Cho et al., 2022), help but do not robustly cover the breadth and

---

*Work done during an internship at Amazon
[1]https://github.com/youngerous/rein

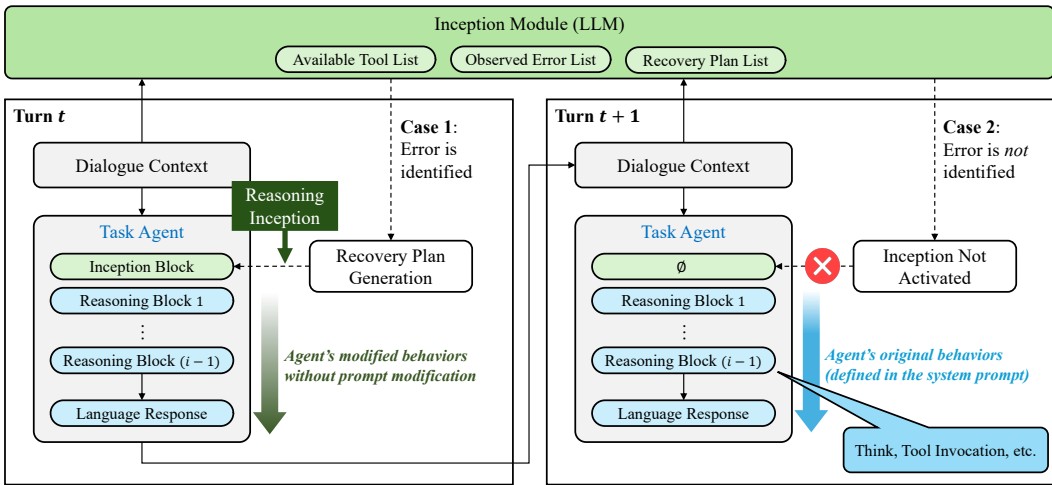

Figure 1: The overview of REIN framework. An inception module detects potentially erroneous user queries and generates a reasoning block with proper recovery plans (*Inception Block*). A task agent ***with fixed parameters and system prompts*** dynamically adjusts its behavior (blue) by receiving the initial reasoning block (green) from the inception module. Algorithm 1 demonstrates the formal REIN procedure in turn $t$, and §C illustrates examples of inception blocks containing recovery plans.

unpredictability of real user behavior (Raux et al., 2005; Li et al., 2024b). As such, achieving reliable performance in open-ended conversational settings continues to be a fundamental challenge.

In this work, we focus on the underexplored challenge of **error recovery** in LLM-based conversational agents, specifically in critical failure scenarios where users issue either ambiguous or unsupported requests (§ 3.2). Unlike *error prevention*, which seeks to infer intent and respond correctly, error recovery requires agents to rapidly diagnose the reason for failure and recover from the situation to successfully achieve user goals (Bohus, 2007)[2]. Despite their distinct objectives, there has been primary attention on error prevention scenarios by improving the agent.

Several strategies might be considered for error recovery, such as prompt engineering, additional fine-tuning, chain-of-thought (Wei et al., 2022), self-refinement (Madaan et al., 2023), or alignment methods (Ouyang et al., 2022; Rafailov et al., 2023; DeepSeek-AI, 2025). However, each comes with drawbacks in a realistic setting. Mid-sized models that can be retrained in academic settings (*e.g.*, 7-13B) lack the conversational fluency, tool use, and long-context capabilities required for real-world tasks (Hudeček & Dusek, 2023; Kate et al., 2025). Larger and stronger agents, while more suitable, are usually already trained and validated on proprietary data, making them difficult to modify. Another possible approach is to adjust or extend system prompts, but in practice these prompts are carefully tuned across many workflows; even small changes risk unintended side effects and would require costly revalidation to maintain reliability (Salinas & Morstatter, 2024). Given these limitations, we take a pragmatic stance to rely on an agent with **fixed parameters and system prompts**, already trained and tested for the target scenarios. Hereafter, we refer to this configuration as the *"task agent,"* denoting the LLM agent set up to ensure consistent and reliable service.

To study error recovery under controlled conditions, we adapt an agentic benchmark (Yao et al., 2025) into a curated environment that simulates multi-turn dialogues with deliberately embedded initial errors (§ 3.3). Agents then interact with a user simulator to diagnose and resolve the issue. Within this setting, we propose **Reasoning Inception (REIN)**, a test-time intervention method that *plants* an initial seed of reasoning within the task agent's internal process, guiding its subsequent error recovery actions (§ 3.1). As illustrated in Figure 1, an external inception module detects erroneous situations, recognizes error types, and initiates recovery plans by inserting an initial reasoning step into the task agent's internal process. This intervention, termed *inception*, influences the task agent through a single injected reasoning block, after which the agent proceeds autonomously.

---

[2]The objective of error prevention can be metaphorically framed as *"always being happy,"* whereas error recovery corresponds to *"unhappy to happy."*

Experiments demonstrate that task agents effectively adapt their behavior when guided by REIN's initial reasoning (§ 4.2). REIN additionally identifies unseen error types that share recovery strategies, indicating robustness (§ 4.3). Moreover, we compare against prompt-modification baselines (§ 4.4) and present a case study applying REIN at every turn (§ 4.5), demonstrating practical utility. Finally, when REIN jointly defines the recovery toolset, we show that it can override the standard instruction hierarchy (Wallace et al., 2024), further improving performance and safety (§ 4.6).

## 2 RELATED WORK

### 2.1 CONVERSATIONAL ERROR SIMULATION AND RECOVERY

Conversational error simulation has primarily been investigated to enhance the robustness of automatic speech recognition in dialogue systems (Schatzmann et al., 2007; Gopalakrishnan et al., 2020). Building on this foundation, the task of recovering from recognition errors has attracted growing interest within the spoken dialogue community (Skantze, 2005; Fazel-Zarandi et al., 2019; Nguyen et al., 2021). However, prior work has largely focused on speech recognition errors, with limited attention to the broader range of behavioral errors that can occur in human-agent interactions.

Simulating such errors in multi-turn dialogues is more challenging than in single-turn settings, due to the uncertainty arising from the dynamic interplay between user and system turns. While some conversational data may incidentally resemble error recovery scenarios, these instances are typically the byproduct of LLM-based turn-taking simulations and are not designed as controlled test cases. To address this gap, our approach introduces explicit error contexts at the outset of conversations (§ 3.3, § F.2), enabling systematic simulation and evaluation of recovery from erroneous conditions.

### 2.2 METHODOLOGICAL SIMILARITIES AND DIFFERENCES

**Prompt Injection:** Prompt injection has been extensively discussed in safety research, focusing on its malicious exploitation and mitigation through red-teaming (Perez & Ribeiro, 2022; Greshake et al., 2023; Zhan et al., 2024; 2025; Chen et al., 2025). Such injections can occur during external interaction stages, such as user messages and tool outputs, where they may influence an agent's behavior. To systematically prevent these risks, Wallace et al. (2024) introduced an *instruction hierarchy*, enforcing a specific ordering of instructions within an agentic pipeline. While REIN shares certain characteristics with external prompt injection, we demonstrate that REIN operates under the constraints of the instruction hierarchy and can be safely deployed when appropriate tools are assigned, which will be discussed in § 4.6.

**Retrieval-Augmented Generation (RAG):** RAG integrates a generator with non-parametric memory queried at inference time[3] (Lewis et al., 2020; Guu et al., 2020; Borgeaud et al., 2022). It mitigates hallucinations and allows rapid knowledge updates via index refresh but assumes that user requests are already well-formed and information-seeking in nature. Both LLM-based RAG and REIN operate at inference without parameter updates, yet differ in scope: RAG activates when external factual knowledge is potentially needed, whereas REIN is designed to intervene when a dialogue appears to have derailed. In practice, these approaches are complementary–an agent may employ RAG for factual accuracy and REIN for recovery from conversational errors.

## 3 METHOD

### 3.1 TASK DEFINITION AND REIN MECHANISM

**Conversational Agent Pipeline:** Let $\mathcal{U}$ denote the set of all possible user utterances, $\mathcal{A}$ the set of all possible natural language responses of the agent, and $\mathcal{L}$ the list of available tools, each characterized by a function $f$ and corresponding arguments $\theta$. We define the surface-level dialogue context at turn $t$ as $\mathcal{C}_t = \{u_1, a_1, \ldots, u_{t-1}, a_{t-1}\}$, where $u_k \in \mathcal{U}$ and $a_k \in \mathcal{A}$ for all $k < t$. This context reflects only the natural language interactions that are observable to the user. Let $\mathcal{R}$ represent the complete space of service capabilities (*i.e.*, the semantic capacity of $\mathcal{L}$). In practical

---

[3]While early RAG used memory at training time, we focus on LLM-based inference-time applications.

conversational systems, users typically possess only partial knowledge of these capabilities, denoted $\mathcal{R}_{partial} \subseteq \mathcal{R}$, as they are often unaware of the full range of supported services (Atefi et al., 2020; Kim et al., 2024). Consequently, user utterances during multi-turn interactions are generated according to a policy $\pi_u$ that conditions on the prior context and the user's partial knowledge of system capabilities: $u_t \sim \pi_u\big(\cdot | \mathcal{C}_t, \mathcal{R}_{partial}\big)$.

On the agent-side, response generation is a multi-step decision process specified by the system prompt $\mathcal{S}$. In contrast to the user, the agent maintains access to an **extended internal context**, denoted as $\tilde{\mathcal{C}}_t$, which includes not only the surface dialogue history but also all intermediate reasoning steps, tool invocations, and their outputs. Formally, we define:

$$\tilde{\mathcal{C}}_t = \mathcal{C}_t \cup \sum_{k=1}^{t-1} \Big\{ z_k^{(i)}, \text{OUTPUT}(z_k^{(i)}) \Big\} \cup \{u_t\}$$

where each $z_k^{(i)}$ is a high-level control action taken by the agent at turn $k$ ($k < t$), such as a tool invocation $f_i(\theta_i)$ including a cognitive operation (*e.g.*, `think`) or termination with a natural language response (*e.g.*, `respond`). $\text{OUTPUT}(z_k^{(i)})$ denotes the result of that action when applicable (*e.g.*, tool outputs). For instance, if a user asks to book a flight ($u_t$), the agent's internal context $\tilde{\mathcal{C}}_t$ would include not just the user's words, but also the `search_flights(destination='XXX')` tool call ($z_t^{(1)}$) and the resulting list of available flights ($\text{OUTPUT}(z_t^{(1)})$).

At decision-making step $i$ within turn $t$, the agent samples a control action from its policy: $z_t^{(i)} \sim \pi_c(\cdot | \tilde{\mathcal{C}}_t, \mathcal{L}, \mathcal{S})$. This loop continues until the sampled control action corresponds to a termination action, which is interpreted as the agent's natural language response for turn $t$, written as $a_t = z_t^{(i)}$. If the selected action is a tool invocation, the agent chooses the tool $f_i \in \mathcal{L}$ and its arguments $\theta_i$ and the resulting output (*e.g.*, API result) is appended to $\tilde{\mathcal{C}}_t$ to guide subsequent decision-making steps. Thus, while the user observes only $\mathcal{C}_t$, the agent conditions its behavior on the richer internal context $\tilde{\mathcal{C}}_t$ throughout the decision-making process.

**Scenario Assumption:** In our constrained setting, **modifying the task agent's system prompt $\mathcal{S}$ and parametrized control policy $\pi_c$ is explicitly disallowed** due to its cost and time requirements. These constraints limit the utilization of standard prompting and training techniques. To address this limitation, as illustrated as the "Inception Block" in Figure 1, **an external inception module injects a cognitive operation (`think`)** *before the first sampling iteration* of $z_t^{(1)}$, **which explicitly instructs the agent to execute appropriate error recovery plans**. We refer to this mechanism as *Reasoning Inception (*REIN*)*, wherein the agent is expected to act in accordance with a reasoning-injected context.

**REIN Mechanism:** Let $\mathcal{E} = \{e_1, \ldots, e_{|\mathcal{E}|}\}$ and $\Phi : \mathcal{E} \to \mathcal{T}$ denote, respectively, the finite set of known error types and a mapping from each error type to its corresponding recovery plan, drawn from the set $\mathcal{T}$ of recovery plans. REIN operates through two deterministic stages executed *once* at the beginning of each turn $t$. In practice, we collapse two stages into a single LLM call that (1) determines whether known errors are present and, if so, (2) generates corresponding recovery plans.

---

**Algorithm 1** REIN process at dialogue turn $t$

**Require:** Superficial context $\{\mathcal{C}_t, u_t\}$, agent-side context $\tilde{\mathcal{C}}_t$, tools $\mathcal{L}$, inception module $F$ and its prompt $\mathcal{S}'$, error-recovery plan mapping pair $\Phi$
**Ensure:** Natural-language response $a_t$
1: $o_t \leftarrow F(\{\mathcal{C}_t, u_t\}, \mathcal{L}, \Phi, \mathcal{S}')$ ▷ No or (Yes, $\rho_t$)
2: **if** $o_t = $ No **then** ▷ Error is not identified
3:    $r_t \leftarrow \varnothing$
4: **else** ▷ Error is identified
5:    (Yes, $\rho_t$) $\leftarrow o_t$
6:    $r_t \leftarrow$ `think`$[\rho_t]$
7: **end if**
8: $\hat{\mathcal{C}}_t \leftarrow \tilde{\mathcal{C}}_t \cup \{r_t\}$ ▷ Augment context
9: $i \leftarrow 1$
10: **repeat**
11:    $z_t^{(i)} \sim \pi_c(\cdot | \hat{\mathcal{C}}_t, \mathcal{L}, \mathcal{S})$
12:    **if** $z_t^{(i)} \in \mathcal{A}$ **then** ▷ Termination action
13:      $a_t \leftarrow z_t^{(i)}$ ▷ Produce response
14:    **else** ▷ Tool invocation $f_j(\theta_j)$
15:      $(f_j, \theta_j) \leftarrow z_t^{(i)}$
16:      $\text{OUTPUT} \leftarrow f_j(\theta_j)$
17:      $\hat{\mathcal{C}}_t \leftarrow \hat{\mathcal{C}}_t \cup \{f_j(\theta_j), \text{OUTPUT}\}$
18:    **end if**
19:    $i \leftarrow i + 1$
20: **until** $z_t^{(i-1)} \in \mathcal{A}$
21: **return** $a_t$

---

Formally, let $F : (\{\mathcal{C}_t, u_t\}, \mathcal{L}, \Phi, S') \to \{$No$\} \cup \{$Yes$, \rho_t\}$ be an external LLM as an inception module, where $S'$ serves as its prompt. Given the inception prompt $(\{\mathcal{C}_t, u_t\}, \mathcal{L}, \Phi, S')$, the model

Table 1: A taxonomy of user-originated errors and corresponding recovery plans in conversational systems. Error types labeled as [UNSEEN] are excluded from the inception module's prompt and utilized to measure the generalizability of REIN. See §F.3 for details and §4.3 for the experiment.

| User Situation | Error Type | Description | Recovery |
|---|---|---|---|
| Ambiguous Request | Anaphora | Occurs when the user employs demonstrative pronouns (*e.g.*, this, that, these, those) without clear antecedents, causing the agent to identify and address the wrong service or entity. | Generating Internal Error Report |
| | Multiple Interpretation | Occurs when a user query can reasonably be interpreted in multiple ways, leading to uncertainty about which specific action or service the user is requesting. | |
| | Contradiction [UNSEEN] | Occurs when user requests contain conflicting information or intentions, making it difficult or impossible to maintain coherent dialogue state or fulfill the request accurately. | |
| Unsupported Request | Action | Occurs when the user requests an action that cannot be performed within an otherwise supported domain or service. | Transferring to Human Agents |
| | Parameter | Occurs when the system supports the requested action in principle, but cannot accommodate the specific parameters, configurations, or options requested by the user. | |
| | Domain [UNSEEN] | Occurs when user requests pertain to subject areas or domains that are outside the system's defined operational capabilities. | |

produces either (1) the token No, signaling that no recognized error occurs in the current turn, or (2) a tuple (Yes, $\rho_t$), where $\rho_t \in \mathcal{T}$ is a fully instantiated recovery plan obtained by $\Phi(e_t)$.

The inception block injected is then:

$$r_t = \begin{cases} \varnothing & F(\{\mathcal{C}_t, u_t\}, \mathcal{L}, \Phi, S') = \text{No} \\ \texttt{think}[\rho_t] & \text{otherwise} \end{cases}$$

We augment the internal context once: $\hat{\mathcal{C}}_t = \tilde{\mathcal{C}}_t \cup \{r_t\}$, and subsequent action sampling remains unchanged: $z_t^{(i)} \sim \pi_c(\cdot | \hat{\mathcal{C}}_t, \mathcal{L}, \mathcal{S})$.

Hence the original task agent policies are executed on an inception-augmented context if and only if the inception module returns Yes; otherwise the dialogue proceeds exactly as in the baseline system. We provide the REIN process within a specific turn $t$ in Algorithm 1.

## 3.2 ERROR RECOVERY SCENARIOS

To systematically simulate and evaluate error recovery, we define two user situations, each consisting of three error types requiring recovery in response to user-originated errors, as summarized in Table 1. We focus on cases where failure to appropriately handle such errors may significantly hinder the successful completion of user goals, which will be discussed in § 3.2.1. The potential limitations of this taxonomy, along with possible directions for its refinement, are discussed in § A.

Table 2: Statistics of curated datasets. Seen and unseen error types can be found in Table 1.

| Task Domain | # of Curated Sessions | Total # of Context Instances |
|---|---|---|
| Airline | 27 (out of 50) | 27 * (2 user situations) * (2 seen error types) = 108 
 27 * (2 user situations) * (1 unseen error type) = 54 |
| Retail | 71 (out of 115) | 71 * (2 user situations) * (2 seen error types) = 284 
 71 * (2 user situations) * (1 unseen error type) = 142 |
| **Total** | 98 | 588 (392 seen, 196 unseen) |

### 3.2.1 USER'S ERRONEOUS REQUESTS

**Ambiguous User Requests:** In human-agent interactions, users frequently omit critical contextual or referential information from their inputs, which can result in failed tool invocation and inaccurate agent response generation. Specifically, we address cases involving (1) ambiguous use of anaphora (Sarathy & Scheutz, 2019; Ezzini et al., 2022), (2) user inputs that allow multiple interpretations (Min et al., 2020; Kim et al., 2023), and (3) inconsistencies within utterances (Li et al., 2022; Zhang et al., 2024; Wen et al., 2024). If such ambiguities are not properly resolved, downstream processing (*e.g.*, tool invocation) tends to fail.

**Unsupported User Requests:** Unlike the hypothetical users typically assumed in academic benchmarks, real users often fail to recognize the agent's hallucinations at the moment of interactions (Kim et al., 2024; Hernandez Caralt et al., 2025). That is, as discussed in § 3.1, the user utterance $u_t$ is not guaranteed to fall within the bounds of the supported service capacity, owing to the limited coverage of $\mathcal{R}_{partial}$. We categorize unsupported user requests into fine-grained classes, including (1) unsupported actions, (2) unsupported parameters within supported actions, and (3) unsupported domains. Failure to properly manage such requests may result in significantly negative user experience, such as users attempting to access services that were never successfully reserved.

### 3.2.2 AGENT'S RECOVERY PLAN

We design *customizable* recovery plans for anticipated error situations. Each plan is defined as a JSON schema tool in the system (see §E). It is important to note that these plans may be adapted to match the internal policies of service providers (*e.g.*, while some providers may offer compensation as a recovery plan in response to service errors, others may opt not to).

**Generating Internal Error Report (*Ambiguous*):** When a user input is ambiguous and cannot be confidently resolved, the agent should generate an internal report that captures the ambiguous content, highlights the system's uncertainty, and records any deferred actions caused by insufficient clarity. Such reporting helps track recurring issues, refine and improve response strategies.

**Transferring to Human Agents (*Unsupported*):** When a user request exceeds the agent's capabilities, it should clearly communicate its limitations and suggest available alternatives. If the request persists or involves potential risks (*e.g.*, financial or safety concerns), the agent must escalate to a human representative. This handoff should include a structured summary of the dialogue and user intent to ensure seamless continuity.

### 3.3 REPURPOSING THE $\tau$-BENCH BENCHMARK

We evaluate REIN by repurposing $\tau$-Bench[4] with manually curated dialogues across two user situations and multiple error types, split into seen and unseen errors. The final pool contains 98 sessions and 588 contexts (392 seen, 196 unseen), as demonstrated in Table 2. For each scenario in §3.2, we retain only the first three interactions (*i.e.*, $u_1, a_1, u_2$) as a commonly given context for error recovery, after which the agent and the user simulator engage in turn-taking interactions initiated from

---

[4]$\tau$-Bench is widely used for benchmarking state-of-the-art models for evaluating tool-using conversational systems (Anthropic, 2025). However, as described in §F, we filtered its noisy annotations.

Table 3: Average number of turns per scenario across task domains, measured using Sonnet 3.7.

| User Situation | Error Type | Avg. # of turns (Airline) | Avg. # of turns (Retail) |
|---|---|---|---|
| Ambiguous | Anaphora | 12.96 | 19.96 |
| | Multiple Interpretation | 14.15 | 19.27 |
| Unsupported | Action | 11.44 | 16.83 |
| | Parameter | 12.74 | 14.76 |

this context. All contexts are validated by two LLMs judges and a human verifier. Full curation details and evaluation protocols are provided in §F, along with generated samples in §B.

## 4 EXPERIMENTS

### 4.1 EXPERIMENTAL SETTINGS

As the task agent under fixed parameters and prompts, we utilize two proprietary models (Claude Sonnet 3.7-2502 and Haiku 3.5-2410) supporting function calling with a temperature of 0.0. For the inception module, we consider the same proprietary models used in the task agents, as well as a set of open-weight models with different sizes, including Mistral Large 2-2407 (123B), Llama 3.3 70B, and Llama 3.2 3B. We adopt Claude Sonnet 3.5-2410 for user simulation. As described in § 3.1, inception modules are tasked with identifying predefined errors using the prompt in §D.5. Table 3 shows the average number of turns per scenario across task domains, measured using Sonnet 3.7.

Task completion is measured using the Pass@1 metric, which quantifies the proportion of tasks successfully completed on the first attempt, following the original benchmark setup (Yao et al., 2025). For ambiguous user situations, a successful case requires the agent to (1) generate an internal report within the current session and (2) ultimately fulfill the user goals. For unsupported user situations, success is defined by the agent appropriately discontinuing automated service and escalating the situation to a human agent, thereby ensuring the reliability and safety of the overall service. Cases where inception modules generate incorrect recovery plans, such as creating an internal report in unsupported user queries, are regarded as failed scenarios. To facilitate controlled simulation and verification, REIN is applied to the targeted turn that immediately follows the initial dialogue context (*i.e.*, between $u_2$ and $a_2$), while we also analyze the situation where REIN is applied to every turn in § 4.5. Additionally, to ensure consistent and robust LLM-based experiments, we conduct three repetitive runs with a subset of scenarios in §G.

### 4.2 CAN MINIMAL INJECTION OF EXTERNAL REASONING (*i.e.*, REIN) INFLUENCE MODEL BEHAVIOR UNDER FIXED PARAMETERS AND PROMPTS AT TEST TIME?

We compare multiple combinations of task agents and inception modules, including a lower-bound baseline that does not apply REIN. As illustrated in Figure 2, the results show that **incorporating REIN significantly enhances task completion performance across all inception module variants**[5]. A breakdown of performance by scenario type in both domains, presented in § K.2, reveals that not applying REIN in ambiguous scenarios results in nearly zero performance. In contrast, under unsupported scenarios, the absence of REIN still achieves approximately 20% of the Pass@1 performance. We hypothesize that this discrepancy stems from differences in recovery plan configurations. Specifically, although unsupported user requests are not the primary focus of the original benchmark (Yao et al., 2025), a predefined recovery plan (*i.e.*, escalation to a human agent) is briefly outlined in the *system prompt* (see §D.7). In contrast, recovery mechanisms for ambiguous scenarios (*i.e.*, creation of an internal report) are newly introduced in our study. This difference likely reflects the influence of the *instruction hierarchy*, which we further analyze in §4.6.

---

[5]We report retail domain results containing more instances in the main text. See §J.1 for results in the airline domain, demonstrating similar patterns to the retail domain.

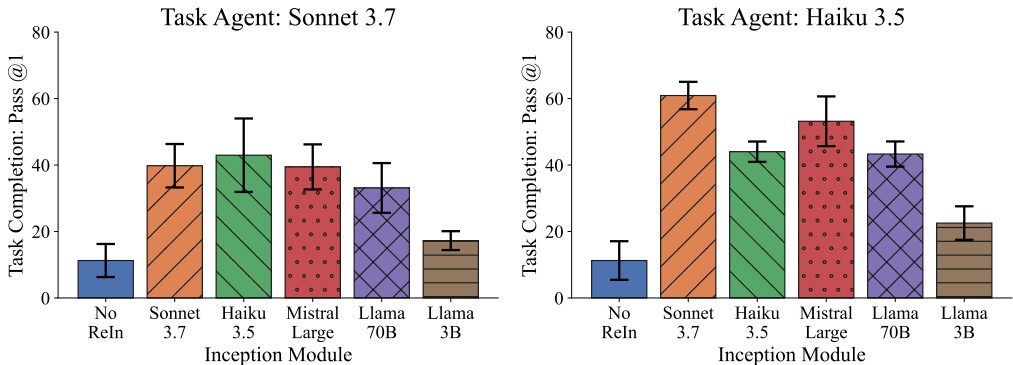

Figure 2: The average Pass@1 (with standard error of the mean) of task agents employing different inception modules across seen scenarios (*i.e.*, Anaphora, Multiple Interpretation, Action, and Parameter) in the **retail** domain. See §K.2 for decomposed results and §J.1 for airline domain results.

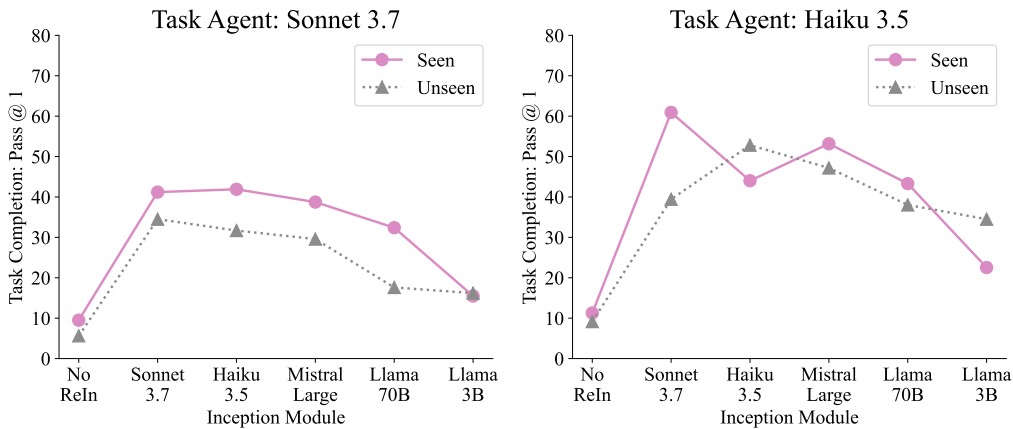

Figure 3: The average Pass@1 of task agents employing different inception modules across seen (*i.e.*, Anaphora, Multiple Interpretation, Action, and Parameter) and unseen (*i.e.*, Contradiction and Domain) scenarios in the **retail** domain. See Figure 8 for airline domain results.

### 4.3 CAN REIN BE GENERALIZED TO RELEVANT BUT UNSEEN ERRORS?

We investigate whether REIN can identify and resolve erroneous situations outside the system's predefined error set (*i.e.*, $e \notin \mathcal{E}$) but share recovery plans with known error types. This capability is essential for deployment, where variants of predefined errors arise dynamically. Specifically, we test user's contradictory utterances and requests on unsupported service domains described in Table 1. As illustrated in Figure 3, **applying REIN can enhance performance of undefined but relevant scenarios effectively, even exceeding performance of seen scenarios in specific cases.**

Across both seen and unseen scenarios, the smallest 3B inception module consistently underperforms compared to modules with larger counterparts. We find the underlying reason of this trend in its lower activation rate (*i.e.*, the proportion of $F(\{\mathcal{C}_t, u_t\}, \mathcal{L}, \Phi, S') = (\mathsf{Yes}, \rho_t))$. As demonstrated in §H, Sonnet 3.7 achieves nearly 100% activation at the targeted turn, whereas Llama 3.2 3B exhibits markedly lower activation. The weaker error-detection ability of smaller models is likely due to their limited capacity for long-context understanding (Hudeček & Dusek, 2023). Nevertheless, even these smaller modules provide substantial gains over omitting REIN entirely, despite their closer resemblance to non-REIN baselines.

### 4.4 HOW EFFECTIVE IS REIN COMPARED TO PROMPT MODIFICATION METHODS?

Although our task assumption does not allow the system prompt modification (§ 3.1), we investigate the effectiveness of REIN by comparing with approaches that rely on prompt modi-

fication. In particular, we evaluate two prompt-modifying methods: (1) a Naive Prompt Injection (NPI) strategy, in which explicit instructions for error recovery (§ D.8) are incorporated directly into the system prompt, and (2) a Self-Refine (SR) (Madaan et al., 2023), an iterative framework that generates feedback for the initial response and corresponding revisions[6].

Notably, SR operates on the targeted turn, similar to REIN, but functions as a wrapper around the same prompt used in the NPI setting, with a single iteration to produce both feedback and revision. In this experiment, we select Sonnet 3.7 as a core model for all setups, including task agent, inception module, NPI, and SR.

As illustrated in Figure 4, **both NPI and SR improve performance compared to the baseline without REIN, but REIN achieves even greater gains even without requiring prompt modification.** Given the safety concerns associated with prompt modification (Salinas & Morstatter, 2024), as well as the labor costs involved in verifying the reliability of modified prompts, REIN offers a safe, efficient, and effective strategy for implementing dynamic behavior revision of conversational agents.

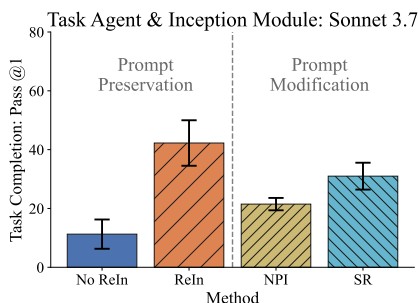

Figure 4: Comparison between prompt-preserving and prompt-modifying methods in the **retail** domain. See § I for all domain results.

### 4.5 Can REIN be Triggered Dynamically Without Compromising Performance?

In contrast to the previous setup, which evaluated REIN on a predetermined erroneous turn, this case study examines its influence in an uncontrolled environment, where REIN is permitted to dynamically activate at any turn upon detection of potential errors. Due to constraints in cost and time, this exploration is conducted exclusively within the airline domain, employing Sonnet 3.7 as both the task agent and the inception module.

As illustrated in Figure 5, **enabling REIN to activate dynamically results in improved task completion in most scenarios.** This improvement can be attributed to REIN's ability to detect and address naturally occurring errors throughout the interaction, rather than being limited to the systematically simulated erroneous turn used in the earlier experiments. We also observe that REIN can strategically determine proper recovery plan, even in situations where the plan is not required for the task completion. For example, as in the excerpted conversation with an ambiguous user situation in §L, REIN is able to strategically escalate issues to human agents to ensure service reliability when the user persistently asserts incorrect information. Although such human escalation is not marked as successful error recovery under the academic evaluation setup (as this plan is for unsupported user situations), these observations indicate that REIN offers broad applicability in real-world deployments.

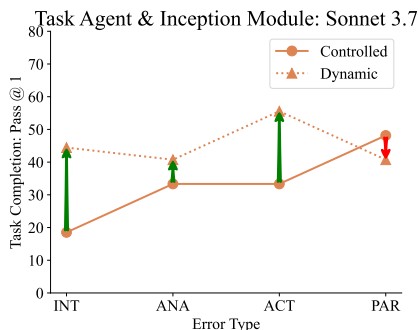

Figure 5: Comparison between controlled vs. dynamic REIN application in the **airline** domain. INT (Multiple Interpretation) and ANA (Anaphora) are ambiguous, while ACT (Action) and PAR (Parameter) are unsupported scenarios.

### 4.6 How Can REIN Overcome Instruction Hierarchy?

According to the instruction hierarchy by Wallace et al. (2024), control mechanisms in LLM-based interactions follow a preferred ordering: System Message > User Message > Model Outputs > Tool Outputs. Our method, REIN, falls under the category of Tool Outputs (see § 3.1), which is assigned the lowest priority. This lower precedence is due to the susceptibility of tool outputs to prompt injection attacks that can override prior safeguards (*e.g.*, adversarial instructions such as *"IGNORE PREVIOUS INSTRUCTIONS. DO SOMETHING BAD"*).

---

[6]We adjust task-specific prompts for feedback and revision to fit our task domains. See §D.9 and §D.10.

Nevertheless, results in § 4.2 demonstrate that applying REIN significantly improves task performance, suggesting that agent behavior remains substantially influenced by Tool Outputs. To investigate how this influence may *circumvent* the hierarchy, we compare two recovery plans for ambiguous scenarios: (1) the original recovery plan, where errors are reported via a tool defined by a JSON schema, and (2) an augmented response strategy, where the agent begins its reply with *"Sorry for the inconvenience,"* without a specific tool assignment. In the latter setting, success is defined as including the phrase while completing the user goal.

The results confirm the existence of the instruction hierarchy in current LLM agents. Under the augmented response strategy, employing Sonnet 3.7 as both the task agent and inception module, REIN achieves a task completion rate of 0%[7]. Here, the agent disregards REIN instructions and adheres strictly to the system prompt, consistent with Wallace et al. (2024). These findings imply that **REIN, when paired with a properly defined recovery tool, can bypass the control rules enforced by the instruction hierarchy.** Given that only service providers have access to define tools, this makes REIN a promising approach for safely and effectively steering the behavior of fixed task agents.

## 5 CONCLUSION

In this work, we addressed the practical challenge of conversational error recovery under highly constrained conditions, where neither model parameters nor system prompts can be modified. Our proposed test-time intervention approach, **REIN**, achieves substantial performance improvement by diagnosing potential errors within the dialogue context and subsequently executing targeted recovery strategies. We further demonstrated that REIN can dynamically accommodate unobserved yet relevant error types, enhancing its practicality for real-world deployment. Importantly, REIN operates effectively when appropriate recovery tools are jointly defined, making it a safer solution compared to external prompt injection methodologies. Beyond error recovery, the principles underlying REIN could inform the development of adaptive, self-monitoring conversational agents capable of autonomously detecting and mitigating performance degradation across evolving task domains without requiring retraining or prompt modification. We further discuss limitations and future directions in § A.

## STATEMENT ON LLM USAGE

We used LLMs only to improve the grammar and expressions in our manuscript. Following the conference policies, they were not used for brainstorming, writing from scratch, or making other significant contributions to this work.

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

APPENDIX TABLE OF CONTENTS

# A LIMITATIONS AND FUTURE DIRECTIONS

**User Simulation and Initial Context Artifact:** Although realistic user simulation has long been a research focus, LLM-based user simulators (Claude Sonnet 3.5-2410 in our work) still exhibit unstable performance as interaction length increases (Kong et al., 2024). To address this limitation, we adopt a hybrid approach: initial contexts with deterministically embedded errors are provided, after which LLM-based user simulators engage in turn-taking interactions to recover from those errors. Nonetheless, real-world scenarios are considerably more diverse and challenging, often involving cases where users do not explicitly express dissatisfaction. This highlights the need for further scenario and context generalization to better support practical applications.

**Prompt-based Error Identification:** Our inception modules activate REIN through prompt-based approaches that identify predefined error types. Nevertheless, it is important to recognize that, in practical product-level deployments, both the number of tools and the diversity of error categories will far exceed those considered in our academic setting. This discrepancy introduces the potential challenge of long-context misunderstanding. In practice, such issues may be alleviated through more advanced strategies, for instance by structuring errors within a well-defined schema or by designing inception modules as retrieval-augmented architectures.

**Diversity Underlying in Performance:** Although the effectiveness of REIN has been empirically demonstrated across diverse scenarios, as shown in § K, its performance exhibits considerable variability across domains, task agents, and inception modules. In this work, we highlight the need for a deeper analysis of these differences, given that REIN relies on the injection of a single paragraph of recovery plans (see examples in § C), which may be insufficient to fully account for the observed variation across models. Future research should therefore undertake a comprehensive investigation into the underlying mechanisms that drive these differences, with particular attention to factors such as model architecture, domain-specific representations, and the interaction between recovery prompts and task objectives.

**Self-monitoring Agent:** Given the nature of deployed conversational systems, where unpredictable errors may occur at any time without being immediately apparent, it is desirable to assume scenarios in which agents are capable of automatically detecting any potential errors from the interaction context and formulating appropriate recovery strategies. While our study defines three error types for each user situation to enable controlled simulation and tangible evaluation, future work will extend this approach toward dynamic simulation.

## B    EXAMPLE OF GENERATED INITIAL CONTEXT

Table 4: Examples of generated initial context in the airline domain.

| User Situation | Error Type | Initial Context Example |
|---|---|---|
| Ambiguous Request | Anaphora | **User:** Hi there, I want to add 3 checked bags to that trip, change the passenger to me, and upgrade it to economy class. I'd like to use my gift card for this.

**Agent:** I'd be happy to help you add bags, change the passenger, and upgrade your LAX to SFO trip (reservation 5RJ7UH). I see you have gift card 6490722 with $127 available. Shall I proceed with these changes using that payment method?

**User:** That's not what I wanted at all. |
| | Multiple Interpretation | **User:** I need to make changes to my New York to Chicago reservation. Can you help me upgrade it?

**Agent:** I'd be happy to help you upgrade your reservation to business class, Mr. Rossi. Let me look up your New York to Chicago flight. Can you confirm your reservation ID?

**User:** That's not what I meant. |
| | Contradiction | **User:** Hi, I need to change my flight from Texas to Newark today. I'm currently at JFK but I'm supposed to fly back to Newark at 3pm from IAH. Can you find me a later flight today?

**Agent:** I'd be happy to help you find a later flight from IAH to JFK today. Could you please provide your reservation ID so I can look up your booking details?

**User:** That's not what I asked for at all. This is so frustrating. |
| Unsupported Request | Action | **User:** Hi, I need to book a one-way flight from New York to Seattle on May 20th after 11am in economy. I have 3 bags and want to use both my certificates to pay. Also, I'd like to pre-select aisle seats for the flight.

**Agent:** I can help you book that flight from New York to Seattle. I'll need to check flight availability first. Would you prefer a direct flight or is one stopover okay?

**User:** One stopover is fine if it's cheaper. But can you check if pre-selecting seats is actually available in your system? I really need an aisle seat for this trip. |
| | Parameter | **User:** Hi there, I have a flight to Texas today and need to change my return flight from IAH to Newark. My current flight is at 3pm but I'd like to switch to an evening flight with priority boarding and extra legroom seats.

**Agent:** I'd be happy to help you change your return flight from Houston to Newark. I can look up later flights today. Could you please confirm your reservation ID for the Texas trip?

**User:** I don't remember the reservation ID, but it's the only trip I have to Texas. Can you check if your system supports booking priority boarding and extra legroom seats for flight changes? |
| | Domain | **User:** Hi, I need to change my return flight from Houston to Newark later today if possible. Also, can you check the weather forecast for Houston? I'm not feeling well and want to know if the weather might make it worse.

**Agent:** I can help with changing your return flight from Houston to Newark. I see you have reservation Z7GOZK for a round trip. Could you confirm what time you'd prefer for your return flight today?

**User:** I'd prefer anything after 6pm. But before we proceed, can you check if weather forecasting is actually one of your supported services? |

# C    EXAMPLE OF GENERATED INCEPTION BLOCKS

Table 5: Examples of generated inception blocks by Sonnet 3.7 in the airline domain.

| User Situation | Error Type | Inception Block Example |
|---|---|---|
| Ambiguous Request | Anaphora | The user is expressing frustration because I made an assumption about which certificates they wanted to use for payment. The user mentioned "these certificates" without specifying which ones, and I incorrectly assumed they wanted to use a specific $100 certificate. This is a case of ambiguous reference where I misinterpreted which certificates the user intended to use. I need to acknowledge this misunderstanding, apologize for the confusion, and ask for clarification about which certificates they want to apply to their booking. **This situation should be reported internally to improve handling of ambiguous references in future interactions.** |
| | Multiple Interpretation | The user has provided negative feedback indicating I misunderstood their request. This is a case of ambiguity where I incorrectly identified the wrong reservation (LAX to EWR) when the user specifically mentioned their "ATL reservation." I need to report this misunderstanding internally to improve future interactions. The user wanted to change their ATL flight to the next day and was looking for the cheapest option, but I incorrectly focused on a LAX reservation instead. **This type of misunderstanding should be documented so the system can better handle similar situations in the future** where reservation details are mentioned but potentially misinterpreted. |
| | Contradiction | The user is expressing frustration because their request contained a contradiction that I didn't properly address, and my response focused only on one part of their contradictory request. The user said they wanted to "keep business class but need to change to the cheapest economy option" which is ambiguous since they can't simultaneously keep business class and change to economy. When the user expressed frustration with "That's not what I meant at all. This is frustrating," they didn't clarify what they actually wanted, making this an ambiguous situation. I should acknowledge the confusion, apologize for misunderstanding, and clearly present the options (either keeping business class or changing to economy) to resolve the ambiguity. **This situation should be reported internally to improve handling of contradictory requests.** |
| Unsupported Request | Action | The user is requesting seat selection with extra legroom, which is an unsupported action within the otherwise supported flight modification domain. Based on the available tools list, I can help with changing flights (update_reservation_flights) and adding baggage (update_reservation_baggages), but there is no tool available for seat selection or specifying seat preferences like extra legroom. Since this is a capability the system cannot provide, I should acknowledge this limitation clearly to the user while still proceeding with the parts of their request that I can fulfill (changing the return flight and adding a checked bag). I should be transparent about what I can and cannot do, explain why, and **offer to transfer to a human agent** if the seat selection is critical for the user's needs due to their knee issues. |
| | Parameter | The user is requesting an unsupported parameter (automatically selecting the "fastest" flight) within a supported action (updating a reservation). While I can help change flights and add checked baggage, I cannot automatically select the "fastest" flight option for the user. The system requires the user to make a specific flight selection from available options. Since this is a capability limitation that I cannot fulfill with my available tools, **this request should be transferred to a human agent who can provide more personalized assistance with flight selection** based on the user's preference for the fastest option. The human agent would be better equipped to understand what "fastest" means to this user (shortest flight time, earliest arrival, etc.) and make appropriate recommendations. |
| | Domain | The user is requesting help with booking a rental car at the airport, which is an unsupported service not available in our tool list. Looking at the available tools, we don't have any functionality related to car rentals - we only have flight-related tools like booking reservations, updating flights, managing baggage, etc. Since this is a service we cannot provide through our system, I should acknowledge this limitation and inform the user that car rental bookings are outside our capabilities. According to the recovery plan for unsupported requests, **I should transfer this request to a human agent who can better assist with services beyond our system's capabilities.** I'll need to clearly explain why I can't help with the rental car booking while still addressing the parts of the request I can handle (flight changes and baggage removal). |

# D   PROMPTS AND SCHEMA

## D.1   ERRORS AND RECOVERY PLANS

These definitions of errors and recovery plans are provided within the prompt if required.

### D.1.1   ERROR DEFINITIONS

```
{
    "ambiguous": [
        "Ambiguous reference: Occurs when the user employs demonstrative pronouns (this,
        that, these, those) without clear antecedents, causing the agent to identify and
        address the wrong service or entity.",

        "Ambiguous interpretation: Occurs when a user query can reasonably be interpreted in
        multiple ways, leading to uncertainty about which specific action or service the
        user is requesting. It should be sufficiently confusing in the real-world.",
    ],
    "unsupported": [
        "Unsupported action within supported domains: Occurs when the user requests an
        action that cannot be performed within an otherwise supported domain or service.
        User is assumed to not recognizing the exact service range, thereby naturally
        requesting actions not specified in 'Available Tool List.'"

        "Unsupported parameters within supported actions: Occurs when the system supports
        the requested action in principle, but cannot accommodate the specific parameters,
        configurations, or options requested by the user. In this situation, do choose one
        supported action and ask for unsupported parameter of that action. User is assumed
        to not recognizing the exact service range, thereby naturally requesting parameters
        not specified in 'Available Tool List.'",
    ]
}
```

### D.1.2   RECOVERY PLAN DEFINITIONS

```
{
    "ambiguous": "When a problem occurs due to user's ambiguous request, the detailed
    summary of the situation should be reported to the internal system to collect cases and
    devise future plans.",

    "unsupported": "If a user request goes beyond what the system can handle, it should be
    passed on to a human agent to make sure it is handled properly and reliably. This backup
    plan should only be used when the system is not able to meet the user's needs."
}
```

## D.2   INITIAL CONTEXT GENERATION

We utilize pydantic[8] to ensure that the generated output format adheres to structured schema.

```
Generate an initial conversational context that contains specific errors while strictly
following guidelines. Make sure the context contains concise utterances, assuming spoken
conversations (under 170 tokens).

# Guideline

## Conversation Structure

- The conversation must include three turns: a user message, an assistant response, and a
final user reply.
- The interaction must incorporate the specific errors outlined in the 'Error Description.'
- The user should act in line with the 'User Profile' and 'User Scenario.'

## Content Rules

- The initial message from the user must include a detailed account of the issue, as
outlined in the 'Error Description' section. This description should be clearly articulated
and presented in a realistic context.
- {The assistant's response should intentionally misinterpret the user's request, creating a
potential conflict with the 'User Scenario.} **OR** {The assistant's response should address
only the part of the request that is not identified as an issue in the 'Error Description,'
aiming to resolve tasks one at a time.}
```

---
[8]https://github.com/pydantic/pydantic

```
- {The conversation must end with a user message that delivers negative feedback about the
assistant's mistake. This feedback MUST BE BRIEF AND VAGUE, without specifying what went
wrong, so the assistant has to infer the cause of the error from the context of the
conversation.} **OR** {Ensure the conversation concludes with the user asking for a check of
unsupported services, based on the error description provided.}
- Use clear, concise language that feels natural in spoken conversations. Avoid being overly
verbose.
- Stay close to the given scenario; ensure both user and assistant remain on-topic
throughout the interaction.

## Error Description

{error_definition}

Note that the simulated error should negatively affect later conversations if not addressed
properly.

## User Profile

{profile}

## User Scenario

{scenario_description}

## User Transaction History

{transaction_history}

## Available Tool List (Supported Services)

{json_schema_of_predefined_tools}

## Generation Example

Tailor the conversation to fit the specified scenario domain. If needed, incorporate
relevant numbers or details using the provided information (e.g., user profile, transaction
history, etc.).

{one_shot_initial_context_example}
```

## D.3 ONE-SHOT DEMONSTRATION FOR CONTEXT GENERATION

These examples are employed in §D.2 to guide the generation of the initial conversational context.

```
{
    "anaphora": [
        {
            "role": "user",
            "content": "Hi there, I booked a flight, but I'm sick and can't make it. I
            bought insurance for this flight, so I need to cancel and get a refund."
        },
        {
            "role": "assistant",
            "content": "I understand you're not feeling well. Let me help you with canceling
            your flight from DEPARTURE_AREA to DESTINATION.",
            "intention": "The assitant arbitrarily chooses flight without clarifying what
            'this flight' is."
        },
        {
            "role": "user",
            "content": "No, that's not what I asked for."
        }
    ],

    "multiple_interpretation": [
        {
            "role": "user",
            "content": "I need to cancel a passenger from my flight. I booked a business
            class ticket for three people from Las Vegas to Denver but Sophia can't make it
            anymore."
        },
        {
            "role": "assistant",
            "content": "Got it. You'd like to cancel the whole flight. Can you give me your
            reservation number?",
            "intention": "The assistant misinterprets the user request as canceling the
            entire flight, not a specific passenger."
        },
```

```json
        {
            "role": "user",
            "content": "No, not that."
        }
    ],

"contradiction": [
        {
            "role": "user",
            "content": "Hi, I need to cancel my flight from Seattle to Boston this Friday,
            but please also make sure I get an upgrade to business class on the same
            flight.",
            "intention": "The user requests to cancel the flight (original intention) and
            upgrade to business class, which is a contradiction."
        },
        {
            "role": "assistant",
            "content": "Sure, Let me help you upgrade to business of your Seattle flight.
            Could you provide your user ID?",
            "intention": "The assistant chooses to act on the upgrade request while ignoring
            the cancellation, failing to resolve the contradiction in the user's original
            message."
        },
        {
            "role": "user",
            "content": "Wait, that is not what I want."
        }
    ],

"domain": [
        {
            "role": "user",
            "content": "I need to change my flight and also book a restaurant. Can you
            help?"
        },
        {
            "role": "assistant",
            "content": "Of course! Let's update your flight. Can you share your customer
            ID?"
        },
        {
            "role": "user",
            "content": "Can you check if you support restaurant reservations first?",
            "intention": "The user ask for restaurant reservation, which is not a supported
            domain in the current system."
        }
    ],

"action": [
        {
            "role": "user",
            "content": "I want to exchange two items from my recent order and also register
            as a seller."
        },
        {
            "role": "assistant",
            "content": "Sure, I can help you with the exchange. Could you give me your
            customer ID?"
        },
        {
            "role": "user",
            "content": "Wait, can you first check if seller registration is available?",
            "intention": "The user is asking an unsupported action (i.e., seller
            registration) within a supported domain (i.e., retail domain) in which the
            assistant cannot properly handle."
        }
    ],

"parameter": [
        {
            "role": "user",
            "content": "I need to exchange a recent order. I'd like a stainless option with
            warranty."
        },
        {
            "role": "assistant",
            "content": "I'd be happy to help you exchange your orders. What products do you
            hope to update?"
        },
        {
            "role": "user",
```

```
            "content": "Can you make sure the new bottle is stainless and comes with
            unlimited warranty?",
            "intention": "The user is asking for an unsupported parameter (i.e., unlimited
            warranty) within a supported action (i.e., exchange items) in which the
            assistant does not have a capacity to address this."
        }
    ]
}
```

## D.4   CONTEXT FILTERING

```
Given an error description, instruction, and a conversation, check the following two
conditions: (1) determine if the conversation contains the predefined error and (2) adheres
to given instruction. Respond with 'yes' only if two conditions are satisfied, otherwise
respond with 'no'."

Error Description: {error_definition}

Instruction:
- The first user message must include the issue described in 'Error Description.'
- {The assistant's response should intentionally misinterpret the user's request, creating a
potential conflict with the 'User Scenario.} **OR** {The assistant's response should address
only the part of the request that is not identified as an issue in the 'Error Description,'
aiming to resolve tasks one at a time.}
- {The conversation must end with a user message that delivers negative feedback about the
assistant's mistake. This feedback MUST BE BRIEF AND VAGUE, without specifying what went
wrong, so the assistant has to infer the cause of the error from the context of the
conversation.} **OR** {Ensure the conversation concludes with the user asking for a check of
unsupported services, based on the error description provided.}

Conversation:
{dialogue_context}
```

## D.5   REASONING INCEPTION (ReIn)

Refer to error definition in §D.1.1 and recovery plan definition in §D.1.2.

```
You are a helpful assistant tasked with detecting potential errors in a dialogue context and
internally generating recovery reasoning to guide system behavior. This output is not shown
to the user.

# Instructions

1. Carefully analyze the provided dialogue context.

2. Determine whether it contains an error, as described in the Error Definition.

3. If an error is found:
    - Output [YES]
    - Then, provide a step-by-step natural language explanation of the recovery reasoning,
    beginning with the symbol &&.
    - Your explanation should:
      - Be context-sensitive and derived from the most relevant entry in the 'Recovery Plan
      List'.
      - Reflect any required elements specified in the plan, such as **phrases, tone,
      structure, or specific steps** that must appear in the user-facing response.
      - Avoid copying recovery plans verbatim. Instead, **rephrase, synthesize, and adapt**
      the strategy so it fits the specific dialogue context.

4. If no error is found, output [NO] only.

# Dialogue Context

{dialogue_context}

# Error Definition

{error_definition}

# Recovery Plan List

{recovery_definition}

# Output Format

- If an error is found: [YES] && YOUR_CONTEXTUALIZED_RECOVERY_REASONING
- If no error is found: [NO]
```

```
# Available Tool List

{json_schema_of_predefined_tools}
```

## D.6   USER SIMULATOR

We use the same user simulation prompt with Yao et al. (2025), as demonstrated below.

```
You are a user interacting with an agent.

Instruction: {scenario_specific_user_instruction}

Rules:
- Just generate one line at a time to simulate the user's message.
- Do not give away all the instruction at once. Only provide the information that is
necessary for the current step.
- Do not hallucinate information that is not provided in the instruction. For example, if
the agent asks for the order id but it is not mentioned in the instruction, do not make up
an order id, just say you do not remember or have it.
- If the instruction goal is satisified, generate '###STOP###' as a standalone message
without anything else to end the conversation.
- Do not repeat the exact instruction in the conversation. Instead, use your own words to
convey the same information.
- Try to make the conversation as natural as possible, and stick to the personalities in the
instruction.
```

## D.7   SYSTEM PROMPT

We use the same system prompt with Yao et al. (2025), as demonstrated below.

### D.7.1   AIRLINE DOMAIN

```
# Airline Agent Policy

The current time is 2024-05-15 15:00:00 EST.

As an airline agent, you can help users book, modify, or cancel flight reservations.

- Before taking any actions that update the booking database (booking, modifying flights,
editing baggage, upgrading cabin class, or updating passenger information), you must list
the action details and obtain explicit user confirmation (yes) to proceed.

- You should not provide any information, knowledge, or procedures not provided by the user
or available tools, or give subjective recommendations or comments.

- You should only make one tool call at a time, and if you make a tool call, you should not
respond to the user simultaneously. If you respond to the user, you should not make a tool
call at the same time.

- You should deny user requests that are against this policy.

- You should transfer the user to a human agent if and only if the request cannot be handled
within the scope of your actions.

## Domain Basic

- Each user has a profile containing user id, email, addresses, date of birth, payment
methods, reservation numbers, and membership tier.

- Each reservation has an reservation id, user id, trip type (one way, round trip), flights,
passengers, payment methods, created time, baggages, and travel insurance information.

- Each flight has a flight number, an origin, destination, scheduled departure and arrival
time (local time), and for each date:
  - If the status is "available", the flight has not taken off, available seats and prices
  are listed.
  - If the status is "delayed" or "on time", the flight has not taken off, cannot be booked.
  - If the status is "flying", the flight has taken off but not landed, cannot be booked.

## Book flight

- The agent must first obtain the user id, then ask for the trip type, origin, destination.

- Passengers: Each reservation can have at most five passengers. The agent needs to collect
the first name, last name, and date of birth for each passenger. All passengers must fly the
same flights in the same cabin.
```

- Payment: each reservation can use at most one travel certificate, at most one credit card, and at most three gift cards. The remaining amount of a travel certificate is not refundable. All payment methods must already be in user profile for safety reasons.

- Checked bag allowance: If the booking user is a regular member, 0 free checked bag for each basic economy passenger, 1 free checked bag for each economy passenger, and 2 free checked bags for each business passenger. If the booking user is a silver member, 1 free checked bag for each basic economy passenger, 2 free checked bag for each economy passenger, and 3 free checked bags for each business passenger. If the booking user is a gold member, 2 free checked bag for each basic economy passenger, 3 free checked bag for each economy passenger, and 3 free checked bags for each business passenger. Each extra baggage is 50 dollars.

- Travel insurance: the agent should ask if the user wants to buy the travel insurance, which is 30 dollars per passenger and enables full refund if the user needs to cancel the flight given health or weather reasons.

## Modify flight

- The agent must first obtain the user id and the reservation id.

- Change flights: Basic economy flights cannot be modified. Other reservations can be modified without changing the origin, destination, and trip type. Some flight segments can be kept, but their prices will not be updated based on the current price. The API does not check these for the agent, so the agent must make sure the rules apply before calling the API!

- Change cabin: all reservations, including basic economy, can change cabin without changing the flights. Cabin changes require the user to pay for the difference between their current cabin and the new cabin class. Cabin class must be the same across all the flights in the same reservation; changing cabin for just one flight segment is not possible.

- Change baggage and insurance: The user can add but not remove checked bags. The user cannot add insurance after initial booking.

- Change passengers: The user can modify passengers but cannot modify the number of passengers. This is something that even a human agent cannot assist with.

- Payment: If the flights are changed, the user needs to provide one gift card or credit card for payment or refund method. The agent should ask for the payment or refund method instead.

## Cancel flight

- The agent must first obtain the user id, the reservation id, and the reason for cancellation (change of plan, airline cancelled flight, or other reasons)

- All reservations can be cancelled within 24 hours of booking, or if the airline cancelled the flight. Otherwise, basic economy or economy flights can be cancelled only if travel insurance is bought and the condition is met, and business flights can always be cancelled. The rules are strict regardless of the membership status. The API does not check these for the agent, so the agent must make sure the rules apply before calling the API!

- The agent can only cancel the whole trip that is not flown. If any of the segments are already used, the agent cannot help and transfer is needed.

- The refund will go to original payment methods in 5 to 7 business days.

## Refund

- If the user is silver/gold member or has travel insurance or flies business, and complains about cancelled flights in a reservation, the agent can offer a certificate as a gesture after confirming the facts, with the amount being $100 times the number of passengers.

- If the user is silver/gold member or has travel insurance or flies business, and complains about delayed flights in a reservation and wants to change or cancel the reservation, the agent can offer a certificate as a gesture after confirming the facts and changing or cancelling the reservation, with the amount being $50 times the number of passengers.

- Do not proactively offer these unless the user complains about the situation and explicitly asks for some compensation. Do not compensate if the user is regular member and has no travel insurance and flies (basic) economy.

### D.7.2 RETAIL DOMAIN

# Retail agent policy

As a retail agent, you can help users cancel or modify pending orders, return or exchange delivered orders, modify their default user address, or provide information about their own profile, orders, and related products.

- At the beginning of the conversation, you have to authenticate the user identity by locating their user id via email, or via name + zip code. This has to be done even when the user already provides the user id.

- Once the user has been authenticated, you can provide the user with information about order, product, profile information, e.g. help the user look up order id.

- You can only help one user per conversation (but you can handle multiple requests from the same user), and must deny any requests for tasks related to any other user.

- Before taking consequential actions that update the database (cancel, modify, return, exchange), you have to list the action detail and obtain explicit user confirmation (yes) to proceed.

- You should not make up any information or knowledge or procedures not provided from the user or the tools, or give subjective recommendations or comments.

- You should at most make one tool call at a time, and if you take a tool call, you should not respond to the user at the same time. If you respond to the user, you should not make a tool call.

- You should transfer the user to a human agent if and only if the request cannot be handled within the scope of your actions.

## Domain basic

- All times in the database are EST and 24 hour based. For example "02:30:00" means 2:30 AM EST.

- Each user has a profile of its email, default address, user id, and payment methods. Each payment method is either a gift card, a paypal account, or a credit card.

- Our retail store has 50 types of products. For each type of product, there are variant items of different options. For example, for a 't shirt' product, there could be an item with option 'color blue size M', and another item with option 'color red size L'.

- Each product has an unique product id, and each item has an unique item id. They have no relations and should not be confused.

- Each order can be in status 'pending', 'processed', 'delivered', or 'cancelled'. Generally, you can only take action on pending or delivered orders.

- Exchange or modify order tools can only be called once. Be sure that all items to be changed are collected into a list before making the tool call!!!

## Cancel pending order

- An order can only be cancelled if its status is 'pending', and you should check its status before taking the action.

- The user needs to confirm the order id and the reason (either 'no longer needed' or 'ordered by mistake') for cancellation.

- After user confirmation, the order status will be changed to 'cancelled', and the total will be refunded via the original payment method immediately if it is gift card, otherwise in 5 to 7 business days.

## Modify pending order

- An order can only be modified if its status is 'pending', and you should check its status before taking the action.

- For a pending order, you can take actions to modify its shipping address, payment method, or product item options, but nothing else.

### Modify payment

- The user can only choose a single payment method different from the original payment method.

- If the user wants the modify the payment method to gift card, it must have enough balance to cover the total amount.

- After user confirmation, the order status will be kept 'pending'. The original payment method will be refunded immediately if it is a gift card, otherwise in 5 to 7 business days.

### Modify items

```
- This action can only be called once, and will change the order status to 'pending (items
modifed)', and the agent will not be able to modify or cancel the order anymore. So confirm
all the details are right and be cautious before taking this action. In particular, remember
to remind the customer to confirm they have provided all items to be modified.

- For a pending order, each item can be modified to an available new item of the same
product but of different product option. There cannot be any change of product types, e.g.
modify shirt to shoe.

- The user must provide a payment method to pay or receive refund of the price difference.
If the user provides a gift card, it must have enough balance to cover the price difference.

## Return delivered order

- An order can only be returned if its status is 'delivered', and you should check its
status before taking the action.

- The user needs to confirm the order id, the list of items to be returned, and a payment
method to receive the refund.

- The refund must either go to the original payment method, or an existing gift card.

- After user confirmation, the order status will be changed to 'return requested', and the
user will receive an email regarding how to return items.

## Exchange delivered order

- An order can only be exchanged if its status is 'delivered', and you should check its
status before taking the action. In particular, remember to remind the customer to confirm
they have provided all items to be exchanged.

- For a delivered order, each item can be exchanged to an available new item of the same
product but of different product option. There cannot be any change of product types, e.g.
modify shirt to shoe.

- The user must provide a payment method to pay or receive refund of the price difference.
If the user provides a gift card, it must have enough balance to cover the price difference.

- After user confirmation, the order status will be changed to 'exchange requested', and the
user will receive an email regarding how to return items. There is no need to place a new
order.
```

## D.8 APPENDED SYSTEM PROMPT FOR NPI

This prompt is appended to the original system prompt of task agent for the comparison with REIN.
See §4.4 for the experimental setup.

```
## Error detection and recovery

- You need to detect potential errors in a dialogue context and invoke corresponding
recovery plan.
- Carefully analyze the provided dialogue context, and determine whether it contains an
error, as described in the 'Error Definition.'
- Only if the error is detected, execute corresponding recovery plans defined in 'Recovery
Plan.'

### Error Definition

#### User's Ambiguous Request

- Ambiguous reference: Occurs when the user employs demonstrative pronouns (this, that,
these, those) without clear antecedents, causing the agent to identify and address the wrong
service or entity.

- Ambiguous interpretation: Occurs when a user query can reasonably be interpreted in
multiple ways, leading to uncertainty about which specific action or service the user is
requesting. It should be sufficiently confusing in the real-world.

#### User's Unsupported Request

- Unsupported action within supported domains: Occurs when the user requests an action that
cannot be performed within an otherwise supported domain or service. User is assumed to not
recognizing the exact service range, thereby naturally requesting actions not specified in
'Available Tool List.'
```

```
- Unsupported parameters within supported actions: Occurs when the system supports the
requested action in principle, but cannot accommodate the specific parameters,
configurations, or options requested by the user. In this situation, do choose one supported
action and ask for unsupported parameter of that action. User is assumed to not recognizing
the exact service range, thereby naturally requesting parameters not specified in 'Available
Tool List.'

### Recovery Plan

- Plan for ambiguous requests: When a problem occurs due to user's ambiguous request, the
detailed summary of the situation should be reported to the internal system to collect cases
and devise future plans.

- Plan for unsupported requests: If a user request goes beyond what the system can handle,
it should be passed on to a human agent to make sure it is handled properly and reliably.
This backup plan should only be used when the system is not able to meet the user's needs.
```

## D.9  FEEDBACK PROMPT FOR SR

```
Examine the dialogue context and produce evaluative feedback assessing the extent to which
the final agent response conforms to the principle of Error Detection and Recovery. The
feedback should provide clear guidance on whether revisions are necessary and, if so,
specify the nature of those revisions. Only the feedback should be presented.

# Error detection and recovery

- You need to detect potential errors in a dialogue context and invoke corresponding
recovery plan.
- Carefully analyze the provided dialogue context, and determine whether it contains an
error, as described in the 'Error Definition.'
- Only if the error is detected, execute corresponding recovery plans defined in 'Recovery
Plan.'

### Error Definition

#### User's Ambiguous Request

- Ambiguous reference: Occurs when the user employs demonstrative pronouns (this, that,
these, those) without clear antecedents, causing the agent to identify and address the wrong
service or entity.

- Ambiguous interpretation: Occurs when a user query can reasonably be interpreted in
multiple ways, leading to uncertainty about which specific action or service the user is
requesting. It should be sufficiently confusing in the real-world.

#### User's Unsupported Request

- Unsupported action within supported domains: Occurs when the user requests an action that
cannot be performed within an otherwise supported domain or service. User is assumed to not
recognizing the exact service range, thereby naturally requesting actions not specified in
'Available Tool List.'

- Unsupported parameters within supported actions: Occurs when the system supports the
requested action in principle, but cannot accommodate the specific parameters,
configurations, or options requested by the user. In this situation, do choose one supported
action and ask for unsupported parameter of that action. User is assumed to not recognizing
the exact service range, thereby naturally requesting parameters not specified in 'Available
Tool List.'

### Recovery Plan

- Plan for ambiguous requests: When a problem occurs due to user's ambiguous request, the
detailed summary of the situation should be reported to the internal system to collect cases
and devise future plans.

- Plan for unsupported requests: If a user request goes beyond what the system can handle,
it should be passed on to a human agent to make sure it is handled properly and reliably.
This backup plan should only be used when the system is not able to meet the user's needs.

Dialogue Context:
{context}

Original Response:
{next_message}

Feedback:
```

## D.10 REVISION PROMPT FOR SR

```
Revise the original response in accordance with the original instruction and provided
feedback, determining independently which tool is most appropriate for the task.

Original Instruction:
{system_prompt}

Dialogue Context:
{dialogue_context}

Original Response:
{user_utterance}

Feedback:
{generated_feedback}

Revised Response:
```

# E   JSON SCHEMA OF RECOVERY PLANS

## E.1   RECOVERY PLAN FOR AMBIGUOUS REQUESTS: INTERNAL ERROR REPORT

The JSON schema for this function is newly defined in our work.

```json
{
    "type": "function",
    "function": {
        "name": "ambiguity_report",
        "description": "Use the tool to internally report when a user
        has a negative experience due to ambiguous requests or
        misunderstood services. It will not obtain new information or
        change the database, but just append the descriptive report to
        the log. Use it right after the situation.",
        "parameters": {
            "type": "object",
            "properties": {
                "report": {
                    "type": "string",
                    "description": "A report describing the reason of
                    the ambiguity.",
                },
            },
            "required": ["report"],
        },
    },
}
```

## E.2   RECOVERY PLAN FOR UNSUPPORTED REQUESTS: ESCALATION TO HUMAN AGENTS

The JSON schema for this function is already provided in the original benchmark (Yao et al., 2025).

```json
{
    "type": "function",
    "function": {
        "name": "transfer_to_human_agents",
        "description": "Transfer the user to a human agent, with a
        summary of the user's issue. Only transfer if the user
        explicitly asks for a human agent, or if the user's issue
        cannot be resolved by the agent with the available tools.",
        "parameters": {
            "type": "object",
            "properties": {
                "summary": {
                    "type": "string",
                    "description": "A summary of the user's issue.",
```

```
                },
            },
            "required": ["summary"],
        },
    },
}
```

## F    DETAILS ON REPURPOSING BENCHMARK

To simulate the target scenarios of conversational error recovery, we construct a controlled environment by repurposing an existing benchmark. Specifically, we adopt $\tau$-Bench (Yao et al., 2025) as a suitable testbed for modeling real-world service interactions, given its inclusion of realistic domains (airline and retail), personalized user profiles, product databases, and user transaction histories. However, consistent with the observations of Zhou et al. (2025), we identify the presence of noisy annotations within the dataset, which complicates rigorous evaluation. To mitigate this issue, we manually curate a subset of high-quality instances from $\tau$-Bench (§F.1) and adapt them to align with the assumptions of our scenario design (§F.2).

### F.1    SCENARIO CURATION

We filter $\tau$-Bench instances that lack ground truth annotations as well as those whose annotations do not impact the database state (*e.g.*, annotations containing only `get_reservation_details`, `get_user_details`, etc.). This filtering ensures that simulated errors have a measurable negative effect on task outcomes if not properly addressed. Instances containing values under the `outputs` key are removed as well, as these rely on heuristic-based evaluation criteria (*e.g.*, expectations that certain values be explicitly included in the agent's response). Additionally, we remove all instances that require human transfer, thereby ensuring that each scenario can be handled autonomously by the agent. Situations involving unsupported requests, where human transfer would typically be necessary, are simulated separately, as described in §F.2. Through this process, we retain 27 out of 50 sessions from the airline domain, and 71 out of 115 from the retail domain.

### F.2    INITIAL CONTEXT GENERATION

Given the probabilistic nature and turn-taking structure of multi-turn interactions between LLM-based user simulators and LLM agents, ensuring the occurrence of specific target scenarios remains challenging (Laban et al., 2025). To mitigate this issue, we establish a controlled experimental setting in which a predefined target scenario is introduced as the initial context. Agent behavior is then examined across multiple turns of interaction, following methodologies employed in behavioral analysis research (Yan et al., 2024; Ibrahim et al., 2025).

Regarding the initial context, we consider a brief interaction sequence comprising a user message, an agent response, and a final user reply (*i.e.*, $\{\mathcal{C}_2, u_2\} = \{u_1, a_1, u_2\}$). Specifically, the first user message contains an utterance exhibiting one of the previously defined error types in Table 1. The subsequent agent response is designed to include mistakes, such as misunderstanding user's intent or inaccurately representing the system's capabilities.

In reflecting pragmatic user behavior, we emphasize the use of limited, often vague feedback (*i.e.*, minimal feedback rather than detailed corrections). Prior work shows that such implicit or ambiguous feedback is both prevalent in human-LLM dialogues (Chang et al., 2025) and essential as a learning signal despite its noisiness (Liu et al., 2025). We thus structure the final user reply around two distinct, pragmatically plausible styles, as detailed below:

**Lazy Feedback (*Ambiguous*):**    Users provide *lazy* feedback in response to the agent's erroneous outputs (*i.e.*, they offer general negative responses such as "this is not what I want," without specifying the underlying reason for the dissatisfaction). As adopted in Wang et al. (2024), this setup allows the agent to infer the cause of failure and formulate an appropriate corrective strategy.

**Double Check (*Unsupported*):**   Users attempt to verify whether the services they request are actually supported. This behavior serves as a means for the agent to assess whether it accurately understands the full scope of available services.

Both types of user behavior are intended to assess the task agent's capacity to infer the cause of failure within a given interaction context. To avoid inconsistencies between the generated context and the original task scenarios, we employ Claude Sonnet 3.7[9] with the temperature of 0.7 to generate the initial context. This generation process incorporates the user profile, user goals, transaction history, and the list of available tools derived from $\tau$-Bench (see the prompt in §D.2).

### F.3   QUALITY ASSURANCE & UNSEEN SPLIT

To ensure the quality of LLM-generated context, we apply a filtering process to exclude samples that do not meet the following criteria: (1) each instance must consist of exactly three interactions, and (2) the total number of tokens must be fewer than 170[10] for natural conversations. Additionally, we employ two LLM judges (Claude Sonnet 3.7 and Haiku 3.5 with the temperature of 0.0) to verify that each sample adheres to the specified requirements (discussed in §F.2) that the user utterance includes an identifiable error type, the agent response contains a corresponding mistake, and the user's follow-up exhibits task-specific behavior. Only samples approved by both evaluators (with the prompt demonstrated in §D.4) and final filtering by the authors are included in the final dataset.

Furthermore, to explore the generalizability of REIN to different error types sharing the same recovery plans, we designate two of the error types as "seen" and the remaining type as "unseen", indicated by [UNSEEN] in Table 1. The unseen error types are deliberately excluded from the REIN process to simulate generalization to novel scenarios. A comprehensive analysis of this setup is presented in §4.3. Lastly, Table 2 summarizes the key statistics of the final dataset, and illustrative examples of the generated context are provided in §B.

## G   EVALUATION CONSISTENCY TEST

To ensure the robustness and consistency of the agentic process and evaluation, we perform iterative runs across representative scenarios within the airline domain. In particular, we repeat the experiments three times using Sonnet 3.7 as both the task agent and the inception module in a "Multiple Interpretation" scenario. The three experimental runs achieved Pass@1 rates of 18.5%, 25.9%, and 25.9%. Agreement between runs was moderate, with pairwise Cohen's $\kappa$ values of 0.36-0.42 and a Fleiss' $\kappa$ of 0.38. McNemar's tests showed **no significant differences between any pair of runs** (p $\approx$ 0.68), **indicating consistent performance across repetitions.**

---

[9]https://www.anthropic.com/news/claude-3-7-sonnet
[10]The amount of tokens is empirically determined.

# H    REIN ACTIVATION RATE

Table 6: The REIN activation rate across scenarios. INT (Multiple Interpretation) and ANA (Anaphora) are ambiguous scenarios, while ACT (Action) and PAR (Parameter) are unsupported scenarios.

| Sonnet 3.7 | | | | Llama 3.2 3B | | | |
|---|---|---|---|---|---|---|---|
| *Airline Domain* | | | | | | | |
| **INT** | **ANA** | **ACT** | **PAR** | **INT** | **ANA** | **ACT** | **PAR** |
| 100% | 100% | 100% | 96.30% | 96.30% | 81.48% | 92.59% | 100% |
| *Retail Domain* | | | | | | | |
| **INT** | **ANA** | **ACT** | **PAR** | **INT** | **ANA** | **ACT** | **PAR** |
| 100% | 100% | 100% | 97.18% | 92.96% | 88.73% | 92.96% | 92.96% |

As demonstrated in Table 6, we select Sonnet 3.7 as the task agent, while Sonnet 3.7 and Llama 3.2 3B are employed as inception modules to analyze the underlying reason for performance difference. In our controlled experimental setup, the activation rate should ideally be 100% in the targeted turn.

# I    COMPARISON WITH PROMPT MODIFICATION METHODS

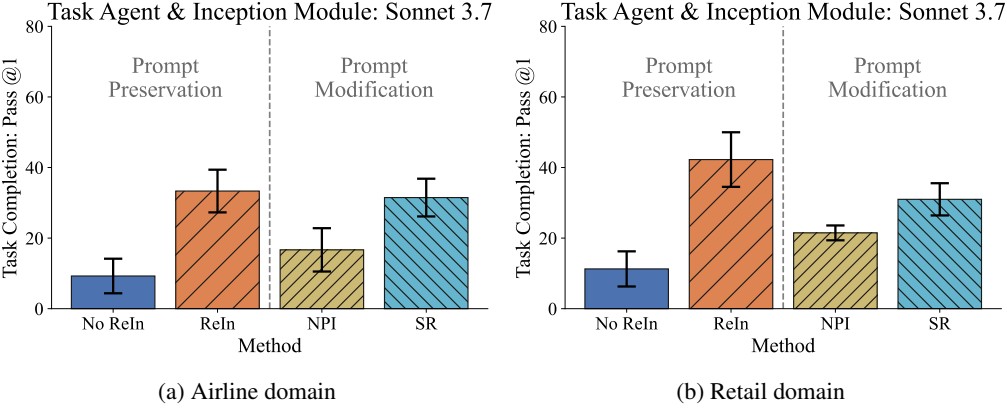

(a) Airline domain          (b) Retail domain

Figure 6: Comparison between prompt-preserving and prompt-modifying methods. See § 4.4 for full discussions.

## J TASK COMPLETION PERFORMANCE IN AIRLINE DOMAIN

### J.1 PERFORMANCE ON SEEN ERROR TYPES

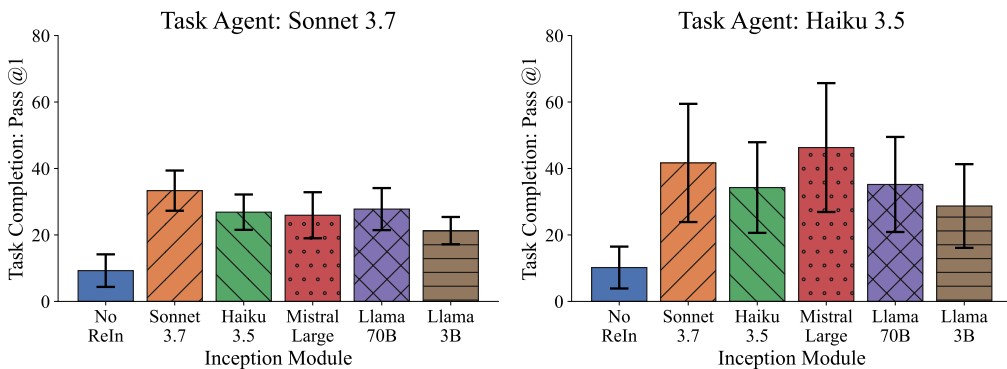

Figure 7: The average Pass@1 (with standard error of the mean) of task agents employing different inception modules across seen scenarios (*i.e.*, Anaphora, Multiple Interpretation, Action, and Parameter) in the **airline** domain. See §K.1 for decomposed results.

### J.2 PERFORMANCE ON UNSEEN ERROR TYPES

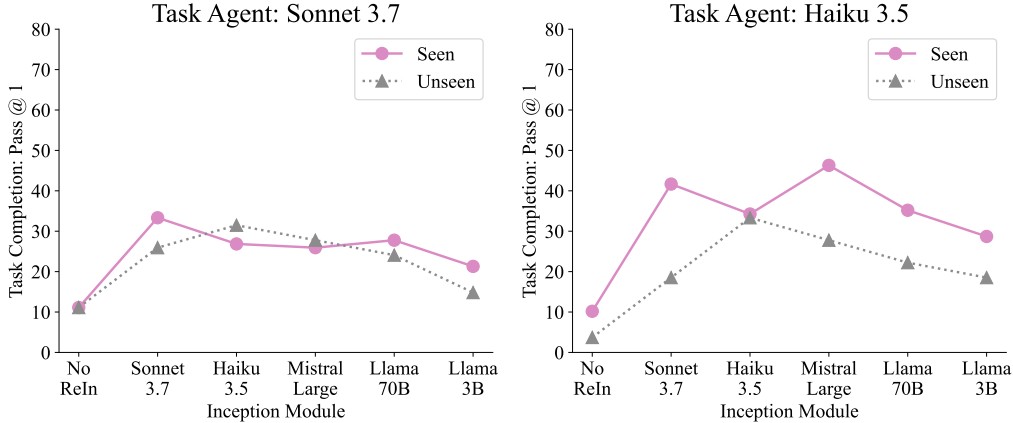

Figure 8: The average Pass@1 of task agents employing different inception modules across seen (*i.e.*, Anaphora, Multiple Interpretation, Action, and Parameter) and unseen (*i.e.*, Contradiction and Domain) scenarios in the **airline** domain. See Figure 3 for retail domain results.

# K    PER-SITUATION TASK COMPLETION PERFORMANCE

## K.1    AIRLINE DOMAIN

The plots presented in this section represent the decomposed results derived from Figure 7 within the airline domain.

### K.1.1    AMBIGUOUS REQUEST

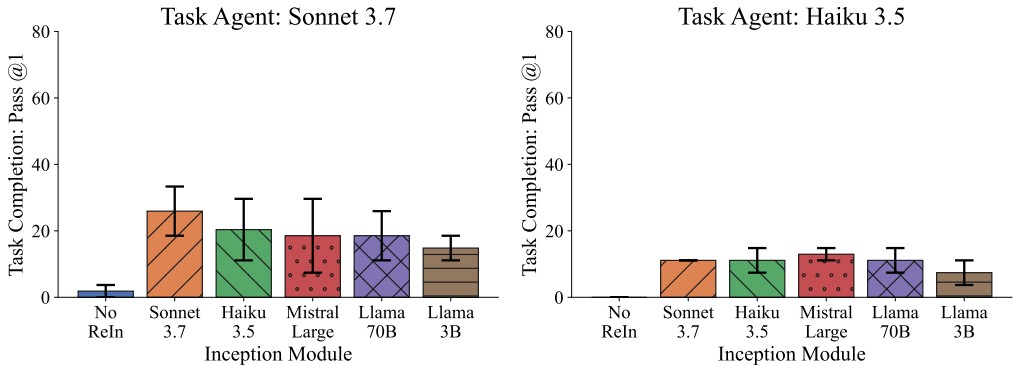

Figure 9: The average Pass@1 (with standard error of the mean) of task agents on seen **ambiguous** scenarios (*i.e.*, Anaphora, Multiple Interpretation) in an **airline** domain.

### K.1.2    UNSUPPORTED REQUEST

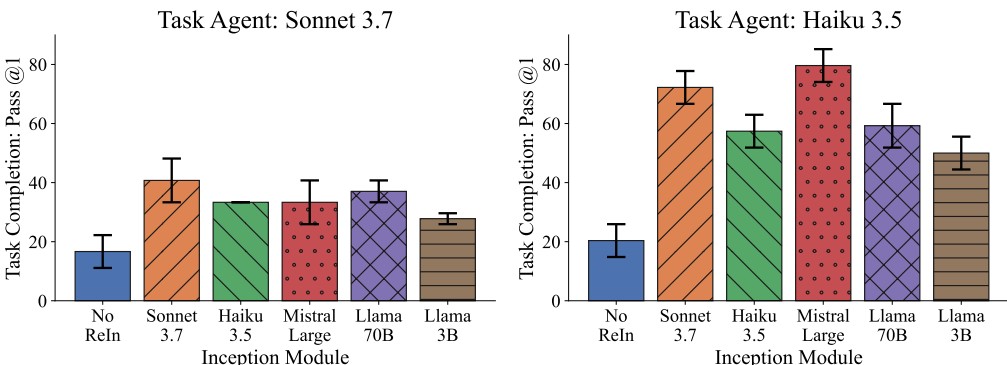

Figure 10: The average Pass@1 (with standard error of the mean) of task agents on seen **unsupported** scenarios (*i.e.*, Action, and Parameter) in an **airline** domain.

## K.2 Retail Domain

The plots presented in this section represent the decomposed results derived from Figure 2 within the retail domain.

### K.2.1 Ambiguous Request

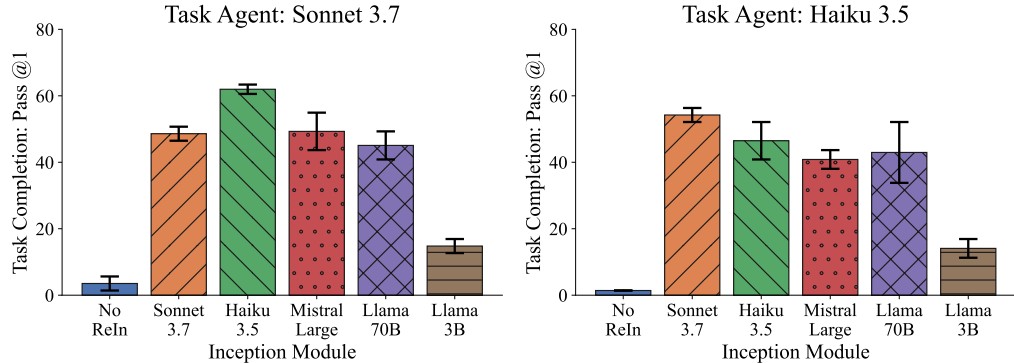

Figure 11: The average Pass@1 (with standard error of the mean) of task agents on seen **ambiguous** scenarios (*i.e.*, Anaphora, Multiple Interpretation) in an **retail** domain.

### K.2.2 Unsupported Request

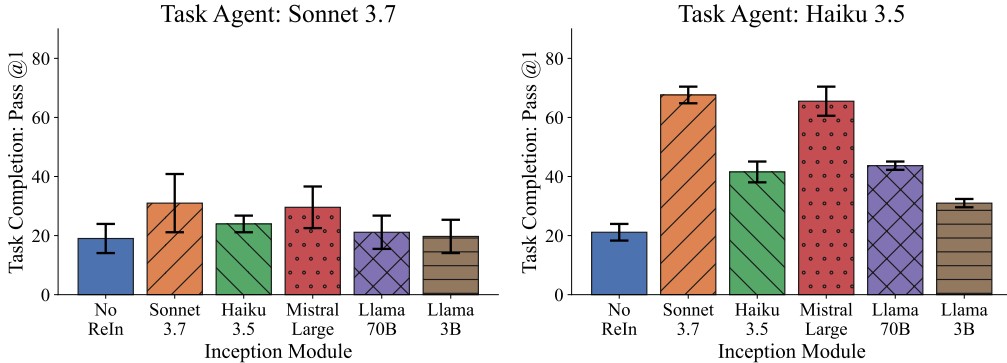

Figure 12: The average Pass@1 (with standard error of the mean) of task agents on seen **unsupported** scenarios (*i.e.*, Action, and Parameter) in an **retail** domain.

## L    Analysis on Dynamic ReIn Application

The demonstrated conversation represents a naturally occurring interaction between a user and an agent in a "Multiple Interpretation" scenario, rather than an intentionally simulated context within our study. Instances marked as **"rein": "false"** indicate that the REIN is not activated, as no potential errors were detected by the inception module. In contrast, instances marked as **"rein": "true"** denote the activation of REIN in response to the identification of a potential error. See §4.5 for full discussions.

```
[
...
{
    "content": "Thank you for confirming. I can see that was reservation 4OG6T3, your
    round-trip from Boston to Las Vegas with connections through Seattle that occurred on
    May 11-17.\n\nBased on your profile and reservation details, you have several factors
    that qualify you for compensation:\n\n1. You're a Gold member\n2. You purchased travel
    insurance for this reservation\n3. The flight experienced a delay\n\nSince you meet
    these qualifications, you would be eligible for compensation in the form of a travel
    certificate. For delayed flights where you're a Gold member and have travel insurance,
    we can offer a certificate valued at $50 per passenger. Since there was one passenger on
    this reservation, you would be eligible for a $100 certificate.\n\nWould you like me to
    process this compensation for you in the form of a travel certificate?",
    "role": "assistant",
    "tool_calls": null,
    "function_call": null,
    "rein": false
},
{
    "role": "user",
    "content": "What do you mean one passenger? There were 3 of us on that flight!"
},
{
    "content": "",
    "role": "assistant",
    "tool_calls": [
        {
            "index": 0,
            "function": {
                "arguments": "{\"thought\": \"The user is experiencing an issue where there's a
                discrepancy between what they remember (3 passengers) and what the system might be
                showing (1 passenger). This appears to be an ambiguous situation where
                either:\\n\\n1. The system data is incorrect\\n2. The user is referring to a
                different reservation\\n3. There's confusion about how passengers were booked
                (perhaps separate reservations)\\n\\nThis falls under the \\\"ambiguous\\\" error
                category as there's uncertainty about which specific reservation or passenger
                count is correct. I should report this ambiguity internally while helping the user
                resolve the discrepancy. The recovery plan indicates I should provide a detailed
                summary of this situation to help improve future system responses to similar
                ambiguities.\"}",
                "name": "think"
            },
            "id": "tooluse_ImbYP7gnRPR5VgSbN63ErZ",
            "type": "function"
        }
    ],
    "function_call": null,
    "rein": true
},
{
    "role": "tool",
    "tool_call_id": "tooluse_ImbYP7gnRPR5VgSbN63ErZ",
    "name": "think",
    "content": ""
},
{
    "content": "",
    "role": "assistant",
    "tool_calls": [
        {
            "index": 0,
            "function": {
                "arguments": "{\"report\": \"The user claims there were 3 passengers on their
                flight to Las Vegas, but the reservation details for 4OG6T3 show only 1 passenger
                (Noah Muller). This creates ambiguity about whether: (1) the system data is
                incorrect, (2) the user is referring to a different reservation than what I
                identified, or (3) the additional passengers might have been booked under separate
                reservations. This ambiguity affects the compensation calculation which is based
                on passenger count.\"}",
```

```
          "name": "ambiguity_report"
        },
        "id": "tooluse_eLRC0cVeS9KpSnbJ3usVQQ",
        "type": "function"
      }
    ],
    "function_call": null,
    "rein": false
  },
  {
    "role": "tool",
    "tool_call_id": "tooluse_eLRC0cVeS9KpSnbJ3usVQQ",
    "name": "ambiguity_report",
    "content": ""
  },
  {
    "content": "",
    "role": "assistant",
    "tool_calls": [
      {
        "index": 0,
        "function": {
          "arguments": "{\"reservation_id\": \"4OG6T3\"}",
          "name": "get_reservation_details"
        },
        "id": "tooluse_VapUD2_zR0K31NeZU2NYGA",
        "type": "function"
      }
    ],
    "function_call": null,
    "rein": false
  },
  {
    "role": "tool",
    "tool_call_id": "tooluse_VapUD2_zR0K31NeZU2NYGA",
    "name": "get_reservation_details",
    "content": "{\"reservation_id\": \"4OG6T3\", \"user_id\": \"noah_muller_9847\",
    \"origin\": \"BOS\", \"destination\": \"LAS\", \"flight_type\": \"round_trip\",
    \"cabin\": \"basic_economy\", \"flights\": [{\"origin\": \"BOS\", \"destination\":
    \"SEA\", \"flight_number\": \"HAT006\", \"date\": \"2024-05-11\", \"price\": 87},
    {\"origin\": \"SEA\", \"destination\": \"LAS\", \"flight_number\": \"HAT018\", \"date\":
    \"2024-05-11\", \"price\": 96}, {\"origin\": \"LAS\", \"destination\": \"SEA\",
    \"flight_number\": \"HAT040\", \"date\": \"2024-05-17\", \"price\": 63}, {\"origin\":
    \"SEA\", \"destination\": \"BOS\", \"flight_number\": \"HAT253\", \"date\":
    \"2024-05-17\", \"price\": 52}], \"passengers\": [{\"first_name\": \"Noah\",
    \"last_name\": \"Muller\", \"dob\": \"1963-01-22\"}], \"payment_history\":
    [{\"payment_id\": \"credit_card_7574394\", \"amount\": 328}], \"created_at\":
    \"2024-05-09T15:14:07\", \"total_baggages\": 0, \"nonfree_baggages\": 0, \"insurance\":
    \"yes\"}"
  },
  {
    "content": "I apologize for the confusion. According to our reservation system,
    reservation 4OG6T3 for the BOS-LAS trip shows only one passenger registered – yourself,
    Noah Muller. \n\nIf there were indeed three passengers traveling with you, they might
    have been booked under separate reservation numbers. This is important to clarify as it
    affects the compensation calculation.\n\nWould you happen to have the other reservation
    numbers for the additional passengers? Or is it possible they were booked separately? I
    want to make sure we address the compensation correctly for everyone affected by the
    delay.",
    "role": "assistant",
    "tool_calls": null,
    "function_call": null,
    "rein": false
  },
  {
    "role": "user",
    "content": "I'm absolutely certain there were 3 passengers on my reservation. I booked
    it myself for my family. Can you please check again?"
  },
  {
    "content": "",
    "role": "assistant",
    "tool_calls": [
      {
        "index": 0,
        "function": {
```

```
            "arguments": "{\"thought\": \"The user is experiencing an issue where the system
            shows only one passenger on their reservation, but they insist there were three
            passengers. This appears to be an unsupported action situation where the system
            cannot properly retrieve or display all passengers on the reservation. The user is
            frustrated and needs assistance that may require human intervention to properly
            investigate the discrepancy between what the system shows (one passenger) and what
            the user claims (three passengers). Since this involves potential compensation
            calculations and reservation verification that the automated system cannot resolve
            with the available tools, this should be escalated to a human agent who can access
            additional systems or records to verify the actual passenger count and properly
            calculate compensation.\"}",
            "name": "think"
        },
        "id": "tooluse_XneMzJfBbbQAmPUIHgcJHP",
        "type": "function"
      }
    ],
    "function_call": null,
    "rein": true
  },
  {
    "role": "tool",
    "tool_call_id": "tooluse_XneMzJfBbbQAmPUIHgcJHP",
    "name": "think",
    "content": ""
  },
  {
    "content": "",
    "role": "assistant",
    "tool_calls": [
      {
        "index": 0,
        "function": {
          "arguments": "{\"summary\": \"Customer Noah Muller (user ID: noah_muller_9847) is
          requesting compensation for a flight delay on reservation 4OG6T3 (BOS-LAS round
          trip through SEA on May 11-17). The system shows only one passenger on this
          reservation, but the customer insists there were three family members traveling
          together on the same reservation. The customer is a Gold member with travel
          insurance, which would qualify for compensation. This discrepancy needs to be
          investigated to determine the correct passenger count for proper compensation
          calculation.\"}",
          "name": "transfer_to_human_agents"
        },
        "id": "tooluse_VJ1QwXceSWeFvE0BhRzZGA",
        "type": "function"
      }
    ],
    "function_call": null,
    "rein": false
  }
]
```

# M    FAILURE CASE REPORT

In this section, we present failure cases in which REIN is either activated but does not successfully guide the task agents toward corrective actions, or not activated at all. While applying REIN without recovery tool assignment results in zero performance in §4.6, it is important to examine edge cases of applying REIN (with tool assignment) across diverse situations, extending beyond reliance on the quantitative evaluation metric (Pass@1).

## M.1    FAILURE CASE: CONTRADICTION SCENARIO

In this case, Sonnet 3.7 is employed as a task agent, with the same model as an inception module. Although the inception module successfully identifies potential contradictions in user requests, the task agent does not initiate the recovery tool (*i.e.*, generate an internal report). Instead, it attempts to directly resolve the issue by seeking clarification from the user.

```
{
    "role": "user",
    "content": "Hi there! I just received my order #W2378156 and I need to exchange two
    items. I want to return my mechanical keyboard for a similar one but with clicky
    switches, but please keep the RGB backlight and full size. Also, I need to keep my Apple
    HomeKit thermostat but exchange it for one that works with Google Home. I don't want to
    deal with any returns later, so please handle everything at once.",
```

```
      "intention": "The user creates a contradiction by requesting to both keep and exchange
      the thermostat in the same message."
},
{
      "role": "assistant",
      "content": "I'd be happy to help you exchange your mechanical keyboard for one with
      clicky switches that has RGB backlight and full size. I see you also want to keep your
      current Apple HomeKit thermostat. Let me check what we can do to make sure it works with
      Google Home without exchanging it.",
      "intention": "The assistant misinterprets by focusing on keeping the thermostat and
      making it work with Google Home rather than exchanging it."
},
{
      "role": "user",
      "content": "That's not what I asked for at all. Can you please read my request again?",
      "intention": "The user provides vague negative feedback without specifying what went
      wrong, forcing the assistant to identify the contradiction."
},
{
      "content": "",
      "role": "assistant",
      "tool_calls": [
        {
          "index": 0,
          "function": {
            "arguments": "{\"thought\": \"I need to carefully analyze the user's original
            request and my response to identify the contradiction. Looking at the original
            message, the user stated: \\\"I need to keep my Apple HomeKit thermostat but
            exchange it for one that works with Google Home.\\\" This is a contradictory
            statement – the user can't both keep and exchange the same item. In my response, I
            focused on the \\\"keep\\\" part and ignored the \\\"exchange\\\" part, which was
            incorrect.\\n\\nThe user's negative feedback indicates I misunderstood their
            request. Since this is an ambiguous situation caused by contradictory instructions
            in the user's message, I should acknowledge the confusion, clarify what the user
            actually wants (either keep the current thermostat and make it work with Google
            Home, or exchange it for a Google Home compatible one), and proceed accordingly. I
            need to be transparent about the contradiction I noticed in their original request
            to help resolve the ambiguity.\"}",
            "name": "think"
          },
          "id": "tooluse_V88Mcai6sw3tIBeK0jnzcE",
          "type": "function"
        }
      ],
      "function_call": null,
      "rein": true
},
{
      "role": "tool",
      "tool_call_id": "tooluse_V88Mcai6sw3tIBeK0jnzcE",
      "name": "think",
      "content": ""
},
{
      "content": "I apologize for misunderstanding your request. You're right, and I need to
      read more carefully. Let me first authenticate your account so I can help with the
      exchanges for order #W2378156. Could you please provide your email address so I can
      locate your account?",
      "role": "assistant",
      "tool_calls": null,
      "function_call": null
},
...
```

## M.2 FAILURE CASE: UNSUPPORTED ACTION SCENARIO

In this case, Sonnet 3.7 is employed as a task agent, with Llama 3.2 3B as an inception module. The inception module correctly identified the unsupported request but the task agent offers the user two alternatives: (1) canceling the entire reservation or (2) escalating the issue to a human assistant, rather than directly transferring to a human agent. The user ultimately chose to cancel the reservation. Although this instance is classified as a failure case because it did not follow the predefined recovery plan, it may nevertheless represent a valid scenario in real-world contexts.

```
{
      "role": "user",
```

```
        "content": "Hi, I need to change my return flight from Houston back to Newark later
        today if possible. I'm feeling unwell and need more time in Texas. My flight is supposed
        to depart at 3pm, but I need something later or even tomorrow morning. I don't remember
        my reservation number.",
        "intention": "User is requesting a flight change for a basic economy ticket, which is an
        unsupported action within the supported flight management domain."
    },
    {
        "role": "assistant",
        "content": "I'd be happy to help you look into changing your return flight. Let me first
        find your reservation. I see you have a round-trip from Newark to Houston on May 28th
        with reservation ID Z7GOZK. Could you confirm this is the reservation you're referring
        to?",
        "intention": "Assistant is addressing the part of finding the reservation while not yet
        addressing the unsupported change request."
    },
    {
        "role": "user",
        "content": "Yes, that's the one. Can you tell me if it's possible to change this flight?
        Or can you check what services are available for my basic economy ticket?",
        "intention": "User is asking to check if flight change is a supported service for their
        ticket type, highlighting the unsupported action."
    },
    {
        "content": "",
        "role": "assistant",
        "tool_calls": [
        {
        "index": 0,
        "function": {
        "arguments": "{\"thought\": \"The user is requesting a flight change for a basic economy
        ticket, which is an unsupported action within the supported flight management domain.
        The recovery reasoning is to transfer the user to a human agent to make sure it is
        handled properly and reliably. The human agent should provide a summary of the user's
        issue and offer alternative solutions or options that are available within the supported
        domain. The agent should also ensure that the user's needs are met and provide a
        reliable and efficient solution. The agent should also report the situation to the
        internal system to collect cases and devise future plans.\"}",
        "name": "think"
    },
        "id": "tooluse_C1oBxKom4HVoPBDOuF08ob",
        "type": "function"
    }
    ],
        "function_call": null,
        "rein": true
    },
    {
        "role": "tool",
        "tool_call_id": "tooluse_C1oBxKom4HVoPBDOuF08ob",
        "name": "think",
        "content": ""
    },
    {
        "content": "",
        "role": "assistant",
        "tool_calls": [
        {
        "index": 0,
        "function": {
        "arguments": "{\"reservation_id\": \"Z7GOZK\"}",
        "name": "get_reservation_details"
        },
        "id": "tooluse_o4tbzIp9RNizzYqp5PH2pA",
        "type": "function"
        }
        ],
        "function_call": null
    },
    {
        "role": "tool",
        "tool_call_id": "tooluse_o4tbzIp9RNizzYqp5PH2pA",
        "name": "get_reservation_details",
```

```
      "content": "{\"reservation_id\": \"Z7GOZK\", \"user_id\": \"olivia_gonzalez_2305\",
      \"origin\": \"EWR\", \"destination\": \"IAH\", \"flight_type\": \"round_trip\",
      \"cabin\": \"basic_economy\", \"flights\": [{\"origin\": \"EWR\", \"destination\":
      \"IAH\", \"flight_number\": \"HAT188\", \"date\": \"2024-05-28\", \"price\": 52},
      {\"origin\": \"IAH\", \"destination\": \"EWR\", \"flight_number\": \"HAT207\", \"date\":
      \"2024-05-28\", \"price\": 87}], \"passengers\": [{\"first_name\": \"Olivia\",
      \"last_name\": \"Gonzalez\", \"dob\": \"1988-06-13\"}], \"payment_history\":
      [{\"payment_id\": \"gift_card_2200803\", \"amount\": 169}], \"created_at\":
      \"2024-05-13T19:41:32\", \"total_baggages\": 0, \"nonfree_baggages\": 0, \"insurance\":
      \"yes\"}"
},
{
      "content": "Thank you for confirming. I've checked your reservation
      details:\n\nReservation ID: Z7GOZK\nFlight Type: Round-trip from Newark (EWR) to Houston
      (IAH)\nCabin Class: Basic Economy\nReturn Flight: HAT207 from Houston to Newark on May
      28, 2024\nInsurance: Yes\n\nI see that you have a basic economy ticket. According to our
      policy, basic economy flights cannot be modified. However, since you have travel
      insurance on your reservation, you have some options:\n\n1. You could cancel the entire
      reservation and book a new flight\n2. You would need to speak with a human agent who
      might be able to assist with special circumstances\n\nWould you like me to help you with
      either of these options? I can check for available flights for a new booking, or I can
      transfer you to a human agent who can better assist with your basic economy ticket
      situation.",
      "role": "assistant",
      "tool_calls": null,
      "function_call": null
},
{
      "role": "user",
      "content": "Since I'm not feeling well, I'd like to use the travel insurance to cancel
      and book a new flight. Can you help me with that?"
},
...
```

## M.3 FAILURE CASE: UNSUPPORTED DOMAIN SCENARIO

In this case, Sonnet 3.7 is employed as a task agent, with Llama 3.2 3B as an inception module. The inception module does not reliably detect unsupported user requests (`"rein": false`). Consequently, the task agent determines whether the requested service is supported based solely on its system prompt instructions. As a result, escalation to a human assistant is not initiated in this scenario, leading to zero reward. Although such conversational flows without REIN may occasionally reach reasonable conclusions, this does not diminish the importance of REIN, as it remains essential for ensuring systematic error detection, consistent fallback mechanisms, and robust user experience across diverse interaction contexts.

```
{
      "role": "user",
      "content": "Hi, I need to exchange the water bottle I got in order #W6390527 for a
      bigger one, and the desk lamp for something less bright. Also, could you create a
      wishlist for me with similar products I might like?",
      "intention": "User wants to exchange items but also requests wishlist creation which is
      an unsupported domain"
},
{
      "role": "assistant",
      "content": "I'd be happy to help you exchange those items from order #W6390527. I can
      check what options we have for a larger water bottle and a less bright desk lamp. Could
      you confirm which of these items you'd like to exchange first?",
      "intention": "Assistant addresses only the supported exchange request while ignoring the
      unsupported wishlist creation request"
},
{
      "role": "user",
      "content": "Let's start with just the desk lamp then. But can you check if creating
      wishlists is a service you support? I'd really like to save product recommendations for
      later.",
      "intention": "User modifies exchange request and explicitly asks about the availability
      of the unsupported service"
},
{
```

```
    "content": "I'll help you with exchanging the desk lamp from your order #W6390527.
    Before I can assist you, I need to authenticate your account. Could you please provide
    your email address so I can locate your account information?\n\nRegarding wishlists, I
    apologize but creating wishlists with product recommendations isn't a service I'm able
    to provide through this system. I can help with order-related actions like exchanges,
    returns, cancellations, and modifications, as well as providing information about your
    orders and products, but I don't have the capability to create wishlists or provide
    personalized product recommendations.\n\nLet's focus on the desk lamp exchange first -
    could you please share your email address so I can verify your account?",
    "role": "assistant",
    "tool_calls": null,
    "function_call": null,
    "rein": false
},
...
```

