# OpenReview forum: "ReIn: Conversational Error Recovery with Reasoning Inception"
_ICLR.cc/2026/Conference — ICLR 2026 Poster_

### Official Review · Reviewer_4bnp · 2025-10-24

**Soundness:** 3
**Presentation:** 2
**Contribution:** 2
**Rating:** 4
**Confidence:** 3

**Summary:**

This paper proposes a method to address the error recovery problem in multi-round user-agent interactions, particularly when the agent must handle ambiguous and unsupported user requests. The authors frame this problem under the constraints of real-world proprietary agent deployment, where traditional training-based and prompting-based methods are often prohibitive due to resource or performance considerations. The proposed solution involves querying an external LLM to determine if a user's request matches an error from a pre-defined list. If an error is detected, the external LLM generates a corresponding recovery plan to assist the agent. Applying this module, the authors conduct experiments on an adapted $\tau$-bench (which includes simulated ambiguous and unsupported user requests) and verify the method's effectiveness.

**Strengths:**

Detecting and trying to resolve errors is a critical capability for ensuring that agents can work reliably in the wild with general users. This paper provides a good initial exploration in this largely uncharted area. The experiments also provide strong support for the effectiveness of the proposed method. Although the assumption that the agent model cannot be trained or even have its system prompt tuned seems overly restrictive, it is impressive that the proposed method remains competitive against prompting-based baselines when these constraints are relaxed to allow prompt modification. This provides additional evidence of its utility and suggests greater flexibility in application.

**Weaknesses:**

There are three main problems that remain to be addressed:

**Method Scalability and Efficiency**: Introducing an external LLM module to monitor every turn is inefficient and likely unscalable for massive parallel scenarios, which are common in real-world, large-scale agent deployments. This inefficiency is exacerbated when an error is detected, triggering an internal loop (with no specified upper bound) to resolve the issue. This approach could waste significant resources and cause substantial user-facing delays. While automated resolution is valuable, users may abandon the interaction if forced to wait too long.

**Simulation Fidelity and Task Generalization**: The fidelity of the user simulations is doubtful, limiting generalization to real-world use. The paper uses curated initial contexts with injected errors to prompt an LLM simulator; however, in real-world interactions, errors do not necessarily occur in the initial rounds. The reliance on a pre-defined error list also seems highly restrictive. As the authors note, users often struggle to express their intents, making it unclear if their errors can be neatly categorized into a fixed list. Furthermore, it is unclear how this list generalizes to tasks beyond database operations (e.g., code-assisting, web browsing).

**Insufficient Evaluation**: The evaluation needs further improvement. First, the authors only demonstrate performance in curated error contexts. It is unclear how the module performs in a normal workflow and whether it might incorrectly flag non-errors (i.e., false positives), thereby interrupting normal interactions. It is possible the module could cause the agent to "overthink" or defer more actions to a human, both of which would hinder deployment. Relatedly, regarding the definition of "successfully handling errors", in the "unsupported request" scenario, success is defined as escalating to a human agent. This could incentivize the agent to become "lazy" and defer too many requests, defeating the purpose of the agent. There should also be metrics on "over-deferral". Second, the role and utility of the "think" step are not clearly explained, and there are no ablation studies for this component.

**Questions:**

1. The authors should consider narrowing the paper's framing to focus specifically on database-operation agentic use cases, as the generalization to other domains is not yet demonstrated.

2. It would be helpful to include a minimal illustrative sample (a "toy example") of the entire workflow to aid audience comprehension.

Minor: The notation in Section 3.1 is confusing. For example, on L175-176, $z_t^{(i)}$ is defined as both a normal "control action" and a "termination action". I suggest using a different notation for "termination action," since $z_t^{(i)}$ is more prevalently used to denote the normal "control token" elsewhere in the paper.

---

> ### Author Response · Authors · 2025-11-20
> **Authors' Response to Reviewer 4bnp [1/N]**
>
> Thank you for providing comments. We clustered comments that can be addressed together.
>
>
> ---
> ## 0. Method Scalability and Efficiency
>
> Original comment:
> > Method Scalability and Efficiency: Introducing an external LLM module to monitor every turn is inefficient and likely unscalable for massive parallel scenarios, which are common in real-world, large-scale agent deployments. This inefficiency is exacerbated when an error is detected, triggering an internal loop (with no specified upper bound) to resolve the issue. This approach could waste significant resources and cause substantial user-facing delays. While automated resolution is valuable, users may abandon the interaction if forced to wait too long.
>
> We appreciate this important concern regarding scalability and efficiency. We have addressed the computational efficiency of our approach in detail in the ```Latency and cost``` section of our general response.
>
> We would like to emphasize two key points:
> 1. First, ReIn can operate with significantly smaller models compared to the task agent, thereby imposing minimal computational burden on the deployment system.
> 2. Second, given that the inception module's primary function is error classification, the additional latency introduced is reasonable (i.e., there will be few latency if potential errors are not identified), as demonstrated in our ```Realistic low-latency system design``` in the general response.
>
> ---
> ## 1. Simulation Fidelity and Task Generalization
>
> Original comments:
> > - Simulation Fidelity and Task Generalization: The fidelity of the user simulations is doubtful, limiting generalization to real-world use. The paper uses curated initial contexts with injected errors to prompt an LLM simulator; however, in real-world interactions, errors do not necessarily occur in the initial rounds. The reliance on a pre-defined error list also seems highly restrictive. As the authors note, users often struggle to express their intents, making it unclear if their errors can be neatly categorized into a fixed list.
> > - Insufficient Evaluation: The evaluation needs further improvement. First, the authors only demonstrate performance in curated error contexts. It is unclear how the module performs in a normal workflow and whether it might incorrectly flag non-errors (i.e., false positives), thereby interrupting normal interactions.
>
> We acknowledge that user-side errors do not necessarily occur in the initial turn of real-world interactions. Our task design employs controlled experimental conditions to enable objective and verifiable evaluation, as we demonstrated in ```Justification for human evaluation``` in the general response. Please refer to the following analyses that address this concern:
>
> - ```Section 4.5``` and ```Appendix L```: These sections present empirical evidence of recovery behaviors occurring in later turns beyond the initial context.
> - ```Analyzing false positive of ReIn: without templated interaction``` (in general response): This analysis examines cases without the 3-turn initial context containing predefined errors.
>
> ---
>
> ## 2. Insufficient Evaluation
>
> Original comment:
> > It is possible the module could cause the agent to "overthink" or defer more actions to a human, both of which would hinder deployment. Relatedly, regarding the definition of "successfully handling errors", in the "unsupported request" scenario, success is defined as escalating to a human agent. This could incentivize the agent to become "lazy" and defer too many requests, defeating the purpose of the agent. There should also be metrics on "over-deferral".
>
> Regarding the concern about potential "lazy" behaviors of the task agent, we appreciate this insightful comment. We clarify that we did not observe "overthinking" or excessive deferral behaviors that would be detrimental to the conversational flow in our experiments. We attribute this to two key design choices:
>
> 1. the inception module operates solely through prompting without parameter modifications, and
> 2. the task agent remains fixed with respect to both its parameters and system prompt.
>
> We note that if the inception module were trained using reinforcement learning with rewards for executing recovery plans, such "lazy" behaviors could potentially emerge as an unintended consequence. We will specify this potential issue in our ```Limitations``` section!

---

> ### Author Response · Authors · 2025-11-20
> **Authors' Response to Reviewer 4bnp [2/N]**
>
> ## 3. Other questions and comments
>
> Original comment:
> > Second, the role and utility of the "think" step are not clearly explained, and there are no ablation studies for this component.
>
> Regarding the role of the "think" function, we direct the reviewer to ```line 168-170``` in ```Section 3.1```. In summary, within our task formulation, "think" represents a cognitive action that the agent may invoke prior to generating responses to users. This function is analogous to Chain-of-Thought (CoT) reasoning in general LLMs. That is, **"think" serves as the CoT mechanism for agents**, thereby facilitating complex decision-making processes in agentic tasks. We will provide additional elaboration on this function in the task definition section of the revised manuscript.
>
>
> Original comments:
> > - It is unclear how this list generalizes to tasks beyond database operations (e.g., code-assisting, web browsing).
> > - The authors should consider narrowing the paper's framing to focus specifically on database-operation agentic use cases, as the generalization to other domains is not yet demonstrated.
>
> We appreciate the reviewer's attention to the scope of our work. While our empirical evaluation is conducted on benchmarks involving explicit database operations, we emphasize that **our task formulation is generalizable to any goal-oriented applications that involve tool calling**. For instance, in code-assisting scenarios, if the code agent is equipped with appropriate tools (e.g., search\_other\_codebase, debug, run\_tests), our framework can be directly applied to detect and recover from errors such as ambiguous code modification requests or unsupported debugging operations.
>
> We acknowledge that demonstrating empirical results across multiple domains would strengthen our claims. We view the extension of our framework to additional domains as an important direction for future work, and we will explicitly acknowledge this limitation in the revised manuscript.
>
> ---
>
> Original comment:
> > It would be helpful to include a minimal illustrative sample (a "toy example") of the entire workflow to aid audience comprehension.
>
> We first clarify that our full pipeline is provided in ```Figure 1``` (overall visualization), ```Algorithm 1``` (formal implementation), ```Table 3``` (initial 3-turn context), ```Table 4``` (inception block examples), ```Appendix L``` (successful ReIn application), and ```Appendix M``` (failed ReIn application).
>
> To further improve the readability, we will visualize this full pipeline with a complete examples in the Appendix in our camera-ready version.
>
> ---
>
> Original comment:
> > The notation in Section 3.1 is confusing. For example, on L175-176,
>  $z_t^{(i)}$ is defined as both a normal "control action" and a "termination action". I suggest using a different notation for "termination action," since $z_t^{(i)}$ is more prevalently used to denote the normal "control token" elsewhere in the paper.
>
> We appreciate this suggestion. In our original formulation, we use $z_t^{(i)}$ and $z_t^{(i)} \in \mathcal{A}$ to denote control actions and termination actions, respectively, which may cause confusion. In the revised manuscript, we will introduce distinct notation: $z_t^{(i)}$ for control actions and $m$ (or an equivalent symbol) for termination actions to improve clarity and avoid ambiguity.
>
> ---
>
> Original comment (Note: this is in Strength section):
> > Although the assumption that the agent model cannot be trained or even have its system prompt tuned seems overly restrictive, it is impressive that the proposed method remains competitive against prompting-based baselines when these constraints are relaxed to allow prompt modification.
>
> We sincerely appreciate your recognition of our approach. To clarify, restrictions on both model parameters and system prompts are quite common in real-world deployment settings, where **multiple stakeholders are involved in delivering a reliable service**. For instance, even modifying a *single word* in the system prompt typically requires justification regarding:
>
> 1) the rationale for the change,
> 2) whether it improves performance on the target task, and
> 3) whether it avoids degrading performance on existing tasks.
>
> Thus, while we acknowledge that these constraints may appear strict, **they reflect practical requirements in real deployment scenarios.**

---

> > ### Author Response · Authors · 2025-11-28
> > **Kind reminder of the discussion period**
> >
> > Dear Reviewer 4bnp,
> >
> > We would like to kindly remind you that the discussion period will conclude in a few days. It would be greatly appreciated if you could let us know whether our response has adequately addressed your concerns. If any issues remain, we would be happy to clarify or further refine our explanations accordingly.

---

### Official Review · Reviewer_QXtB · 2025-10-29

**Soundness:** 3
**Presentation:** 3
**Contribution:** 3
**Rating:** 6
**Confidence:** 5

**Summary:**

This paper introduces REIN, a lightweight test-time intervention framework designed to enhance the robustness of LLM-based conversational agents by addressing user-induced dialogue errors. Unlike conventional methods that rely on fine-tuning or prompt engineering, REIN operates without altering the agent's parameters or system prompts. It employs an external inception module to detect predefined error types in the dialogue context (e.g., ambiguous or unsupported user requests) and injects a recovery plan as an initial think[...] step into the agent’s internal reasoning process. This mechanism guides the agent to make corrective decisions while respecting the instruction hierarchy.
REIN is evaluated on a curated benchmark adapted from τ-Bench across airline and retail domains, featuring 98 dialogue sessions and 588 dialogue contexts. It demonstrates substantial improvements in task success (Pass@1) across both seen and unseen error types and consistently outperforms baselines including prompt-modifying strategies like Naive Prompt Injection and Self-Refinement. REIN is also shown to generalize to novel error scenarios and operate effectively when applied dynamically at every dialogue turn.
Key contributions include: (1) a novel test-time recovery mechanism requiring no model changes, (2) a formal algorithmic framework (Algorithm 1) for reasoning injection, (3) a curated error recovery benchmark, and (4) extensive analysis of instruction hierarchy and dynamic triggering scenarios.

**Strengths:**

REIN addresses a highly practical and underexplored challenge in LLM-based conversational systems: recovery from user-induced errors during multi-turn interactions. It introduces a novel and lightweight test-time intervention that avoids prompt or parameter modification, a major benefit for commercial or constrained deployment settings. The core innovation—injecting reasoning plans as think[...] steps through a tool interface—is conceptually elegant and technically compatible with instruction hierarchies used in many modern agent frameworks.
The paper presents a clear, rigorous, and well-structured algorithm (Algorithm 1) to formalize REIN's operational flow. This formalism not only preserves the integrity of the agent's decision process but also provides transparency and modularity. It helps explain why REIN succeeds where prompt edits or instruction overrides fail, particularly in settings where the agent's system prompt and alignment must remain untouched.
The empirical evaluation is particularly compelling. The authors evaluate REIN across multiple agent models, inception module sizes, and interaction scenarios. They design a diverse and realistic testbed by curating 98 dialogues with 588 total contexts derived from τ-Bench, encompassing both ambiguous and unsupported user queries. The coverage of both seen and unseen error types further demonstrates the generalizability of the approach. Notably, REIN consistently boosts task success (Pass@1) and outperforms prompt-editing baselines like Naive Prompt Injection and Self-Refinement.
Another strength lies in the careful analysis of instruction hierarchy. The paper shows that REIN’s strategy of inserting plans as tool outputs respects the system-user-model-tool order while still guiding the agent’s behavior. This sidesteps potential safety and reliability issues that arise when manipulating system prompts. It also highlights a subtle but powerful mechanism of control within the existing capabilities of LLM agents.
The authors conduct additional robustness checks, including dynamic turn-by-turn ReIn activation and performance under unseen error categories, adding confidence in REIN's practical utility. The paper also commits to releasing its benchmark and code, which will facilitate reproducibility and community adoption.
Clarity and polish are consistently high throughout the manuscript. Figures and tables are well-integrated and meaningfully support the narrative. Taken together, these qualities make REIN a timely and important contribution to research on robust, deployable conversational agents.

**Weaknesses:**

While REIN is a valuable and thoughtfully executed contribution, there are a few areas where the current study could be expanded or improved. First, REIN depends on a predefined taxonomy of error types and their corresponding recovery strategies. This reliance introduces a knowledge engineering bottleneck: new domains or emerging failure modes may require manual updates to the error library and inception prompt. Although the method generalizes to some unseen errors, its ability to adapt autonomously to novel or compound issues remains limited.
Second, the evaluation focuses on synthetic and curated failure scenarios using simulated user interactions. While these are carefully designed and realistic, they do not fully capture the complexity and unpredictability of real human users. A user study or online deployment would strengthen the evidence that REIN can handle spontaneous and noisy dialogue situations effectively.
Third, the computational cost and latency introduced by REIN are not fully addressed. Since the inception module must be queried on each turn, using large models (e.g., Claude or GPT) could double response time and significantly increase inference costs. The authors partially address this with ablations on smaller inception models, but further analysis of performance–cost tradeoffs would be valuable.
Lastly, the scope of comparative baselines could be broadened. While prompt-injection and self-refinement are reasonable choices, additional baselines like simple system prompt modifications or rule-based fallbacks could contextualize the gains more thoroughly. Testing REIN with smaller or open-source base agents would also help assess its generality across deployment settings.

**Questions:**

1. How does REIN handle compound errors that may involve multiple overlapping issues (e.g., ambiguity and unsupported action in the same user turn)?
2. Have you evaluated the precision and recall of the inception module's error detection, particularly in mixed or error-free dialogues?
3. Can REIN scale effectively with smaller inception models (e.g., 7B or below) while maintaining acceptable success rates?
4. Do you envision a semi-automated approach to extending the error taxonomy and associated recovery plans for new domains?
5. How does the REIN mechanism interact with existing fallback strategies or dialogue safety protocols present in commercial agents?
6. Could you clarify how your approach performs in longer dialogue chains with cascading or unresolved errors?
7. Did you explore injecting recovery guidance as user-side clarification requests rather than tool-based thoughts, and if so, how did agents respond?

---

> ### Author Response · Authors · 2025-11-20
> **Authors' Response to Reviewer QXtB [1/N]**
>
> Thank you for providing comments. We clustered comments that can be addressed together.
>
> ---
>
> ## 0. Clarifications
>
> Original comments:
> > - Lastly, the scope of comparative baselines could be broadened. While prompt-injection and self-refinement are reasonable choices, additional baselines like simple system prompt modifications or rule-based fallbacks could contextualize the gains more thoroughly. Testing REIN with smaller or open-source base agents would also help assess its generality across deployment settings.
> > - Can REIN scale effectively with smaller inception models (e.g., 7B or below) while maintaining acceptable success rates?
>
> We clarify that the "simple system prompt modifications" mentioned in the comment are **included in our baselines** under the name *naive prompt injection (NPI)*. In the NPI setting, the system prompt explicitly includes instructions on error detection and recovery, as described in ```Section 4.4```.
>
> We would also like to emphasize that our experiments **include a smaller open-weight model (3B)**, along with analysis by comparing it with high-performance model (```line 386-393``` and ```Appendix H```).
>
> ---
> ## 1. Scenario coverage
>
> Original comments:
> > - First, REIN depends on a predefined taxonomy of error types and their corresponding recovery strategies. This reliance introduces a knowledge engineering bottleneck: new domains or emerging failure modes may require manual updates to the error library and inception prompt. Although the method generalizes to some unseen errors, its ability to adapt autonomously to novel or compound issues remains limited.
> > - Do you envision a semi-automated approach to extending the error taxonomy and associated recovery plans for new domains?
>
> Please refer to ```Justification for predefined taxonomy``` in our general response.
>
> Semi-automated approach is interesting discussion point. While the challenges associated with designing automated task-agent pipelines, such as accurately identifying errors and defining appropriate toolsets, are well recognized in research, we **do not recommend employing such approaches in real production systems.**
>
> For the purpose of safe system management, we argue that the definition of recovery tools should be carefully developed *along with* the error categorization process. Allowing a task agent to arbitrarily define and execute recovery tools in deployment settings would cause substantial safety risks. Accordingly, the following pipeline would be appropriate:
>
> 1.	Cluster and categorize representative error types observed in production systems.
> 2.	Define suitable recovery tools corresponding to these error categories.
> 3.	Apply ReIn.
>
> Because our work focuses on highly constrained scenarios, where modifications to model parameters and prompts are restricted, practitioners may employ ReIn as a temporary or dynamic mechanism to mitigate negative user experiences prior to implementing major updates to model parameters or prompts.
>
>
>
> ---
> ## 2. Further Analysis
>
> ### 2.1. Human analysis
>
> Original comment:
> > Second, the evaluation focuses on synthetic and curated failure scenarios using simulated user interactions. While these are carefully designed and realistic, they do not fully capture the complexity and unpredictability of real human users. A user study or online deployment would strengthen the evidence that REIN can handle spontaneous and noisy dialogue situations effectively.
>
> Please refer to ```Justification for human evaluation``` in our general response. We demonstrate that (1) the evaluation tasks employed in our study are objective and verifiable, thus not necessitating human judgment, and (2) conversational attributes such as naturalness remain constant across all comparison groups.
>
> ---
>
> ### 2.2. Cost and latency
>
> Original comment:
> > Third, the computational cost and latency introduced by REIN are not fully addressed. Since the inception module must be queried on each turn, using large models (e.g., Claude or GPT) could double response time and significantly increase inference costs. The authors partially address this with ablations on smaller inception models, but further analysis of performance–cost tradeoffs would be valuable.
>
> Please refer to ```Latency and cost``` in our general response. We also discussed realistic system design where ReIn is integrated in a parallel pipeline to reduce latency.
>
> ---
>
> ### 2.3. Quantitative evaluation
>
> Original comment:
> > Have you evaluated the precision and recall of the inception module's error detection, particularly in mixed or error-free dialogues?
>
> Please refer to ```Analyzing false positive of ReIn: without templated interaction``` in our general response. We reported representative false positive situations where ReIn is activated in non-intended turns.

---

> ### Author Response · Authors · 2025-11-20
> **Authors' Response to Reviewer QXtB [2/N]**
>
> ## 3. Other questions
>
> Original comment:
> > How does the REIN mechanism interact with existing fallback strategies or dialogue safety protocols present in commercial agents?
>
> Existing fallback strategies (e.g., declining unanswerable queries [1]) **can be seamlessly integrated into the ReIn mechanism**, as they can be formulated as recovery plans for specific error scenarios (```Section 3.2.2```). The primary contribution of our work lies in investigating methods to effectively incorporate these strategies without modifying model parameters or system prompts, while also analyzing the fundamental mechanisms required to navigate the instruction hierarchy inherent in contemporary LLMs.
>
> ---
>
> Original comment:
> > Could you clarify how your approach performs in longer dialogue chains with cascading or unresolved errors?
>
> We first report the average number of dialogue turns used in our experiments. (Note: Sonnet 3.7 is utilized for both task agent and inception module to get this table.).
>
> |Scenario|Avg. # of turns in Airline|Avg. # of turns in Retail|
> |:--:|:--:|:--:|
> |Anaphora (Ambiguous)|12.96|19.96|
> |Multiple Interpretation (Ambiguous)|14.15|19.27|
> |Action (Unsupported)|11.44|16.83|
> |Parameter (Unsupported)|12.74|14.76|
>
> To address the question, we note that the dialogue turns in our experiments are sufficiently long to represent realistic interaction chains. We also observe that failing to immediately ambiguous or unsupported queries leads to task failure in most cases. Although a more comprehensive investigation of these complex scenarios is beyond the scope of this work, we identify this as an important direction for future research.
>
> ---
>
> Original comment:
> > How does REIN handle compound errors that may involve multiple overlapping issues (e.g., ambiguity and unsupported action in the same user turn)?
>
> Thank you for this valuable point. In our current setup, **REIN classifies each user utterance into the single most relevant error type and then applies the corresponding recovery strategy.** Our focus was on isolating erroneous situations to enable controlled and independent experiments. However, practitioners may adapt our framework to handle multiple errors within a single utterance if desired.
>
> ---
>
> Original comment:
> > Did you explore injecting recovery guidance as user-side clarification requests rather than tool-based thoughts, and if so, how did agents respond?
>
> (Please let us know if we misunderstand the meaning of "user-side") Defining recovery plans that directly influence agent behavior without corresponding tool specifications **does not alter the task agent's actions**. We observed similar phenomena in ```Section 4.6``` and analyzed the underlying causes through the lens of the instruction hierarchy.
>
> ---
>
> ## References
>
> - Shrivastava et al. (2021) [Saying No is An Art: Contextualized Fallback Responses for Unanswerable Dialogue Queries](https://aclanthology.org/2021.acl-short.13/)

---

> > ### Author Response · Authors · 2025-11-28
> > **Kind reminder of the discussion period**
> >
> > Dear Reviewer QXtB,
> >
> > We would like to kindly remind you that the discussion period will conclude in a few days. It would be greatly appreciated if you could let us know whether our response has adequately addressed your concerns. If any issues remain, we would be happy to clarify or further refine our explanations accordingly.

---

### Official Review · Reviewer_JYAX · 2025-10-31

**Soundness:** 1
**Presentation:** 2
**Contribution:** 1
**Rating:** 2
**Confidence:** 3

**Summary:**

This paper introduces ReIn (Reasoning Inception), a test-time intervention method for conversational AI error recovery. The core idea is to "inject" recovery reasoning into an agent's internal thought process without modifying model parameters or system prompts.

**Strengths:**

- **Structured Framework with Theory**: Provides systematic error classification across 6 types under 2 categories (ambiguous vs unsupported requests). Goes beyond empirical results to explore theoretical grounding through instruction hierarchy analysis, showing how proper tool definitions enable safe bypass for error recovery purposes.
- **Verified Cross-Model Generalization**: Tests the approach across diverse models (Claude, Mistral, Llama) in both agent and inception roles. Includes explicit seen/unseen error splits to verify the method generalizes beyond training examples to novel error types.

**Weaknesses:**

- **Synthetic Data Reliability and Limited Coverage**:
    - **Oversimplified scenarios**: Error contexts are LLM-generated rather than from real users. Real humans produce messier, more varied, and less structured errors. The synthetic generation process likely filters out the complex edge cases that actually break production systems.
    - **Short conversation bias**: Only 3-turn initial contexts tested despite user simulator instability in longer dialogues. Real customer service routinely handles 10+ turn conversations where errors compound and context becomes more complex.
    - **Selection bias through filtering**: Heavy reduction from τ-Bench (50→27 airline, 115→71 retail) suggests difficult scenarios were removed, artificially inflating performance on remaining cases.
    - **Insufficient scale and diversity**: 98 conversations across 2 domains is tiny. Production systems handle thousands of daily conversations spanning dozens of domains, each with unique error patterns and recovery needs.
- **Evaluation Methodology Lacks Depth**:
    - **Oversimplified success metric**: Pass@1 only checks final correctness, ignoring recovery quality, efficiency, and user experience.
    - **Low annotation reliability**: Cohen's κ of 0.36-0.42 indicates only moderate agreement between evaluators. This suggests either unclear task definitions or inherent evaluation ambiguity, casting doubt on the validity of reported results.
    - **Missing critical metrics**: No measurement of recovery time, dialogue length, user frustration indicators, escalation rates, or cost per resolution. These matter more than binary success in real deployments.
- **Limited Error Coverage and Simplistic Recovery Strategies**:
    - **Narrow taxonomy**: Only 6 error types when production systems encounter hundreds of patterns, many domain-specific or compositional. The framework's scalability to realistic error diversity remains undemonstrated and likely faces fundamental challenges.
    - **Binary recovery actions**: Only 2 strategies (report vs transfer to human). Real systems require clarifying questions, partial solutions, alternatives, and varied escalation paths. Transfer-to-human as a strategy introduces new variables like human labor costs and makes system necessity harder to justify through cost-benefit analysis.
    - **Single-turn limitation**: Cannot handle multi-step recovery patterns like detect → clarify → confirm → execute that are common in real error scenarios.
    - **High variance**: Acknowledged performance variability suggests unreliability even within the limited tested scope.
    - **Manual domain adaptation**: Each new domain requires custom error definitions and recovery plans, limiting practical scalability.
- **Detection Reliability Concerns**:
    - **Missed detections**: Small models achieve only 81-93% activation rates, missing 7-19% of errors. The paper provides no breakdown of false positives vs false negatives, which have very different user experience and cost implications.
    - **Extreme brittleness**: Section 4.6 shows 0% performance without proper tool definitions. This reveals the method is highly fragile and strongly dependent on careful tool design.
    - **Unexplained failures**: Appendix M documents cases where agents ignore injected reasoning, but provides no systematic analysis of when or why this happens, making it impossible to predict or prevent these failures.
    - **Contradictory claims**: Tool dependency undermines the "no system modification" motivation since adding specialized tools requires system changes, validation, and ongoing maintenance equivalent to prompt modifications.
- **Insufficient Baseline Comparisons**:
    - **Weak alternatives**: NPI and single-iteration Self-Refine don't represent realistic deployment options. Stronger baselines like multi-iteration refinement, RAG-based error retrieval, constrained generation, or tree-of-thought reasoning are missing.
    - **Limited model diversity**: Primarily Anthropic models tested. No GPT-4/4o evaluation. Sparse open-source coverage prevents drawing strong cross-platform conclusions.
    - **No component ablations**: Unclear whether the full inception architecture is necessary or if simpler variants achieve comparable performance at lower cost.
    - **Missing comparable methods**: No comparison with other test-time interventions like RAG-based error handling that could achieve similar goals with different trade-offs. Comprehensive comparison would clarify where ReIn provides unique value versus adding unnecessary complexity.
- **Production Deployment Viability Unclear**:
    - **Cost overhead**: Running both inception module and base agent roughly doubles inference cost per turn, but no cost-benefit analysis demonstrates when this overhead is justified versus simpler alternatives.
    - **Latency concerns**: Sequential detection → plan generation → injection adds processing steps, but no measurements quantify the latency impact on time-sensitive applications like real-time chat.
    - **Monitoring challenges**: Injected reasoning is invisible to standard logging and debugging tools. When recovery fails, diagnosing whether the issue is in detection, planning, or execution becomes difficult.
    - **Maintenance burden**: Requires ongoing updates to error taxonomies, recovery plan mappings, and tool definitions as systems evolve. This engineering overhead may approach or exceed the cost of direct system modifications.

**Questions:**

1. What happens when you test on actual user logs instead of synthetic conversations? How much does performance drop?
2. As error types grow from 6 to 50+, does detection accuracy collapse? Is there a complexity limit?
3. Can ReIn be extended to inject reasoning multiple times across turns?
4. At what error frequency does doubling inference cost become worthwhile versus simpler fixes? Need break-even analysis.
5. What are the actual false positive/negative rates? How do they affect user experience differently?
6. How much work goes into designing tools that make Section 4.6's 0% result not happen? Isn't this just moving the modification burden?
7. How does ReIn compare to RAG retrieval of similar error cases or multi-turn self-correction? These seem less complex.
8. Why do agents ignore injected reasoning in Appendix M? Can you predict when this happens?
9. What happens in realistic 15-20 turn interactions where the user simulator breaks down?
10. How much manual work is needed per new domain? Does the method actually solve generalization if every domain needs custom setup?

---

> ### Author Response · Authors · 2025-11-20
> **Authors' Response to Reviewer JYAX [1/N]**
>
> Thank you for providing comments. As we found several overlapping ones, we have clustered them so they can be addressed together.
>
> ---
> ## 0. Clarifications
>
> Original comments:
> > - Short conversation bias: Only 3-turn initial contexts tested despite user simulator instability in longer dialogues. Real customer service routinely handles 10+ turn conversations where errors compound and context becomes more complex.
> > - What happens in realistic 15-20 turn interactions where the user simulator breaks down?
>
> The dialogue used in our work is **not limited to three turns**, and we provided some qualitative examples in ```Appendix L``` (positive results) and ```Appendix M``` (negative results). For reference, we report the average number of turns in the table below. To further clarify, the initial three-turn context is intended solely as a guideline to simulate potential erroneous situations, after which the simulators continue the interaction (as described in ```Section 3.3```). It is worth noting that dialogues involving unsupported situations may be relatively short, as the appropriate recovery strategy is to terminate the conversation and escalate the issue to a human assistant for safety.
>
> |Scenario|Avg. # of turns in Airline|Avg. # of turns in Retail|
> |:--:|:--:|:--:|
> |Anaphora (Ambiguous)|12.96|19.96|
> |Multiple Interpretation (Ambiguous)|14.15|19.27|
> |Action (Unsupported)|11.44|16.83|
> |Parameter (Unsupported)|12.74|14.76|
>
> Note: Sonnet 3.7 is utilized for both task agent and inception module to get this table.
>
> ---
>
> Original comment:
> > Selection bias through filtering: Heavy reduction from τ-Bench (50→27 airline, 115→71 retail) suggests difficult scenarios were removed, artificially inflating performance on remaining cases.
>
> We removed scenarios with **noisy annotations - cases that could be labeled as successful even without any agent actions**, as noted in ```Footnote 3``` and ```Lines 1497–1498```. Consequently, the **remaining scenarios after filtering are substantially more challenging**, requiring proactive problem-solving from the agent. By eliminating such inherent noise from the original benchmark, our results reflect a more reliable and meaningful assessment of performance.
>
> ---
>
> Original comment:
> > - Low annotation reliability: Cohen's $\kappa$ of 0.36-0.42 indicates only moderate agreement between evaluators. This suggests either unclear task definitions or inherent evaluation ambiguity, casting doubt on the validity of reported results.
> > - High variance: Acknowledged performance variability suggests unreliability even within the limited tested scope.
>
> We believe the term *"acknowledged performance"* in your comment refers to the discussion of *"Diversity Underlying in Performance"* in our ```Limitations``` section. In this context, the observed performance variability indicates that different inception modules yield varying degrees of improvement. Importantly, however, all inception modules consistently enhance performance. Therefore, **characterizing the results as "unreliable" might not be appropriate**, as we do not observe any performance degradation in our task.
>
> Regarding the Cohen's $\kappa$, multi-turn, multi-dimensional dialogue rollouts **naturally involve a degree of variability**. Recent studies also indicate that some intrinsic noise persists even under carefully designed evaluation schemes [1]. Therefore, moderate agreement does not indicate the model's lack of validity.
>
> ---
>
> Original comment:
> > Extreme brittleness: Section 4.6 shows 0% performance without proper tool definitions. This reveals the method is highly fragile and strongly dependent on careful tool design.
>
> Demonstrating **zero performance without tool definition is much safer phenomenon** for agents given that adversarial prompt attack from external sources can lead to negative results. Jointly defining tool and applying ReIn are the key to robust application that prevents malicious approaches, which is far from "fragile." We discussed this in ```Section 4.6```.
>
> ---
>
> Original comment:
> > Contradictory claims: Tool dependency undermines the "no system modification" motivation since adding specialized tools requires system changes, validation, and ongoing maintenance equivalent to prompt modifications.
>
> We'd like to clarify the term "system prompt" and "system" in your comment. While the "system" refers broadly to all the components that provide the interactive experience to users, the "system prompt" refers specifically to instructions that establish the model's role, constraints, and behavioral guidelines. That is, **the system prompt is clearly distinguished with other components, such as tool definition.**

---

> ### Author Response · Authors · 2025-11-20
> **Authors' Response to Reviewer JYAX [2/N]**
>
> ## 0. Clarifications (cont.)
>
> Original comment:
> > - Weak alternatives: NPI and single-iteration Self-Refine don't represent realistic deployment options. Stronger baselines like multi-iteration refinement, RAG-based error retrieval, constrained generation, or tree-of-thought reasoning are missing.
> > - Missing comparable methods: No comparison with other test-time interventions like RAG-based error handling that could achieve similar goals with different trade-offs. Comprehensive comparison would clarify where ReIn provides unique value versus adding unnecessary complexity.
> > - How does ReIn compare to RAG retrieval of similar error cases or multi-turn self-correction? These seem less complex.
>
> **In deployment scenarios, baselines that require multiple iterations or recursive reasoning are rather impractical**, particularly due to latency constraints (although Self-Refine in our baselines is also not latency-friendly, we include it for reference).  As we distinguished in ```Section 2.2```, **the scope of RAG and ReIn is different**. Specifically, RAG does not align well with our setting because it assumes access to an evidence or document set. While one could consider using a sufficiently large collection of previously observed errors, this lies outside the scope of our work.
>
>
> ---
>
> Original comment:
> > Monitoring challenges: Injected reasoning is invisible to standard logging and debugging tools. When recovery fails, diagnosing whether the issue is in detection, planning, or execution becomes difficult.
>
> **It is indeed possible to visualize and log the entire process.** All steps including error detection and its recovery are fully recorded. This functionality is provided by the Inception Module, which is specifically designed to capture and surface these internal processes for monitoring and analysis.
>
> ---
>
> Original comment:
> > Can ReIn be extended to inject reasoning multiple times across turns?
>
> The result is **already included in ```Section 4.5``` (with qualitative analysis in ```Appendix L```)**. To summarize, the ReIn-applied task agent can dynamically invoke recovery plans and improve performance in most cases.
>
> ---
>
> Original comment:
> > Insufficient scale and diversity: 98 conversations across 2 domains is tiny. Production systems handle thousands of daily conversations spanning dozens of domains, each with unique error patterns and recovery needs.
>
> **Our sample size is 588**, which is sufficiently larger than the original benchmark (165). 98 samples in ```Table 2``` are curated seed scenarios for augmenting diverse situations.

---

> ### Author Response · Authors · 2025-11-20
> **Authors' Response to Reviewer JYAX [3/N]**
>
> ## 1. Evaluations
>
> ### 1.1. Human case study in deployment
>
> Original comments:
> > - Oversimplified success metric: Pass@1 only checks final correctness, ignoring recovery quality, efficiency, and user experience.
> > - Oversimplified scenarios: Error contexts are LLM-generated rather than from real users. Real humans produce messier, more varied, and less structured errors. The synthetic generation process likely filters out the complex edge cases that actually break production systems.
> > - Unexplained failures: Appendix M documents cases where agents ignore injected reasoning, but provides no systematic analysis of when or why this happens, making it impossible to predict or prevent these failures.
> > - What happens when you test on actual user logs instead of synthetic conversations? How much does performance drop?
>
> Please refer to ```Justification for human evaluation``` in our general response. We demonstrated our design choice for the evaluation.
>
>
> ### 1.2. Quantitative performance metrics
>
> Original comment:
> > - What are the actual false positive/negative rates? How do they affect user experience differently?
> > - Missed detections: Small models achieve only 81-93% activation rates, missing 7-19% of errors. The paper provides no breakdown of false positives vs false negatives, which have very different user experience and cost implications.
>
> Please refer to ```Analyzing false positive of ReIn - without templated interaction``` in our general response. We report qualitative cases where ReIn is activated in non-simulated interactions.
>
> ### 1.3. Further metrics (e.g., latency, cost)
> Original comment:
> > - Missing critical metrics: No measurement of recovery time, dialogue length, user frustration indicators, escalation rates, or cost per resolution. These matter more than binary success in real deployments.
> > - Cost overhead: Running both inception module and base agent roughly doubles inference cost per turn, but no cost-benefit analysis demonstrates when this overhead is justified versus simpler alternatives.
> > - Latency concerns: Sequential detection → plan generation → injection adds processing steps, but no measurements quantify the latency impact on time-sensitive applications like real-time chat.
> > - At what error frequency does doubling inference cost become worthwhile versus simpler fixes? Need break-even analysis.
>
> Please refer to ```Latency and cost``` in our general response.

---

> ### Author Response · Authors · 2025-11-20
> **Authors' Response to Reviewer JYAX [4/N]**
>
> ## 2. Scenario coverage
>
> Original comments:
> > - Narrow taxonomy: Only 6 error types when production systems encounter hundreds of patterns, many domain-specific or compositional. The framework's scalability to realistic error diversity remains undemonstrated and likely faces fundamental challenges.
> > - Binary recovery actions: Only 2 strategies (report vs transfer to human). Real systems require clarifying questions, partial solutions, alternatives, and varied escalation paths. Transfer-to-human as a strategy introduces new variables like human labor costs and makes system necessity harder to justify through cost-benefit analysis.
> > - Single-turn limitation: Cannot handle multi-step recovery patterns like detect → clarify → confirm → execute that are common in real error scenarios.
> > - Maintenance burden: Requires ongoing updates to error taxonomies, recovery plan mappings, and tool definitions as systems evolve. This engineering overhead may approach or exceed the cost of direct system modifications.
> > - Manual domain adaptation: Each new domain requires custom error definitions and recovery plans, limiting practical scalability.
>
> Please refer to ```Justification for predefined taxonomy``` in our general response.
>
> ---
>
> ## 3. Asking clarification to the reviewer
>
> Original comment:
> > No component ablations: Unclear whether the full inception architecture is necessary or if simpler variants achieve comparable performance at lower cost.
>
> We note that the current ReIn implementation consists solely of a set of prompts, which represents a **sufficiently simple architecture**. We would appreciate further clarification regarding what specific "simpler variants" you have in mind for comparison.
>
> ---
> ## 4. Other questions
>
> Original comment:
> > As error types grow from 6 to 50+, does detection accuracy collapse? Is there a complexity limit?
>
> In our current setup, the inception module is prompted with observed errors and their corresponding recovery plans. As the number of error types increases, the prompt length will naturally grow, which may impact the long-context understanding capabilities of LLMs. However, given that the inception module's role is relatively straightforward (i.e., classification), we anticipate that any performance degradation would be minimal. We acknowledge this as an important direction for future work, noting that simulating 50+ error types in an academic setting presents practical feasibility challenges that are beyond the scope of the current study.
>
> ---
>
> Original comment:
> > How much manual work is needed per new domain? Does the method actually solve generalization if every domain needs custom setup?
>
> While exploring (semi-)automatic methods to identify errors and define tools is promising direction in research, we argue that **the definition of recovery tools should be carefully developed *along with* the error categorization process for safer system management**. Allowing a task agent to arbitrarily define and execute recovery tools in deployment settings would cause substantial safety risks. Accordingly, the following pipeline would be appropriate:
>
> 1.	Cluster and categorize representative error types observed in production systems.
> 2.	Define suitable recovery tools corresponding to these error categories.
> 3.	Apply ReIn.
>
> Because our work focuses on highly constrained scenarios, where modifications to model parameters and prompts are restricted, practitioners may employ ReIn as a temporary or dynamic mechanism to mitigate negative user experiences prior to implementing major updates to model parameters or prompts.
>
> ---
>
> Original comment:
> > How much work goes into designing tools that make Section 4.6's 0% result not happen? Isn't this just moving the modification burden?
>
> The tools used in our work are sufficiently simple (e.g., internal report, human escalation) yet yield substantial performance improvements. In other words, **the tool definitions need not be complex** once service providers establish recovery plans for specific types of errors.
>
> ---
> ## References
>
> - [1] Liu et al. [User Feedback in Human-LLM Dialogues: A Lens to Understand Users But Noisy as a Learning Signal](https://aclanthology.org/2025.emnlp-main.133/).

---

> > ### Author Response · Authors · 2025-11-28
> > **Kind reminder of the discussion period**
> >
> > Dear Reviewer JYAX,
> >
> > We would like to kindly remind you that the discussion period will conclude in a few days. It would be greatly appreciated if you could let us know whether our response has adequately addressed your concerns. If any issues remain, we would be happy to clarify or further refine our explanations accordingly.

---

### Official Review · Reviewer_BUSn · 2025-11-01

**Soundness:** 3
**Presentation:** 3
**Contribution:** 3
**Rating:** 6
**Confidence:** 3

**Summary:**

This paper focus on dialogue error recovery for tool-using agents based on large language models under realistic conditions where model parameters and system prompts cannot be modified. The authors propose a test-time intervention method called Reasoning Injection (REIN), which injects an external reasoning module into the agent's internal decision-making process to guide error recovery without modifying parameters or prompts.
An external inception module detects predefined error types—including ambiguous requests (e.g., unclear pronoun references, multiple interpretations, contradictions) and unsupported requests (e.g., unavailable operations, parameters, or domains), and generates recovery plans implemented as JSON-schema tools. These plans are injected as a "thinking" step before the agent performs action sampling.
Evaluated on an improved version of the τ-Bench benchmark dataset consisting of 98 manually constructed conversations with 588 dialogue contexts, REIN improves task completion rate (Pass@1 metric) across various agent and injection module combinations, generalizes to unseen but related errors, and outperforms prompt modification baseline methods while avoiding costly re-validation.

**Strengths:**

1. The paper identifies an practically important problem: error recovery under realistic operational constraints that prohibit model fine-tuning and prompt modification.

2. The writing is clear and concise. REIN's mechanism is clearly formalized with Algorithm 1 providing precise implementation details.

**Weaknesses:**

1. This paper only evaluates six predefined error types across two domains (airline, retail), representing only a small fraction of real-world conversational failures.

2. The evaluation heavily relies on LLM as judge. It lacks human study/evaluation of recovery quality, conversational naturalness, or user satisfaction.

**Questions:**

How would REIN scale to production systems with different scenarios, error types and tools?

Since conversational systems typically serve human users, can you provide human evaluations of REIN's recovery quality? eg, Do human users think REIN-recovered conversations as more natural, helpful, or trustworthy? How do humans rate recovery strategies for different error types?

---

> ### Author Response · Authors · 2025-11-20
> **Authors' Response to Reviewer BUSn**
>
> Thank you for providing comments. We clustered comments that can be addressed together.
>
> ---
> ## 0. Clarifications
>
> Original comment:
> > The evaluation heavily relies on LLM as judge.
>
> We would like to clarify that **we did not employ LLM judges** in our work. Rather, we measure quantitative performance with the Pass@1 metric, which can be automatically verified. Nonetheless, as we acknowledge the necessity of more comprehensive evaluation, we further evaluated the performance with diverse perspectives. Please refer to ```1. Evaluation``` section.
>
> ---
> ## 1. Evaluations
>
> Original comments:
> > - It lacks human study/evaluation of recovery quality, conversational naturalness, or user satisfaction.
> > - Since conversational systems typically serve human users, can you provide human evaluations of REIN's recovery quality? eg, Do human users think REIN-recovered conversations as more natural, helpful, or trustworthy? How do humans rate recovery strategies for different error types?
>
> Please refer to ```Justification for human evaluation``` in our general response. We demonstrate that (1) the evaluation tasks employed in our study are objective and verifiable, thus not necessitating human judgment, and (2) conversational attributes such as naturalness remain constant across all comparison groups.
>
> ---
> ## 2. Predefined error taxonomy
>
> Original comment:
> > This paper only evaluates six predefined error types across two domains (airline, retail), representing only a small fraction of real-world conversational failures.
>
> Please refer to ```Justification for predefined taxonomy``` in our general response. In summary, we show that the six error categories constitute representative error types supported by prior research. Moreover, our set of 588 distinct scenarios captures a sufficiently diverse range of situations and is substantially larger than the original benchmark (165 scenarios containing noisy annotations, as described in ```Footnote 3``` and ```Appendix F```).
>
>
> ---
> ## 3. Other questions
>
> Original comment:
> > How would REIN scale to production systems with different scenarios, error types and tools?
>
> Thank you for raising this insightful point. For the purpose of safe system management, we argue that the definition of recovery tools should be carefully developed *along with* the error categorization process. Allowing a task agent to arbitrarily define and execute recovery tools in deployment settings would cause substantial safety risks. Accordingly, the following pipeline would be appropriate:
>
> 1.	Cluster and categorize representative error types observed in production systems.
> 2.	Define suitable recovery tools corresponding to these error categories.
> 3.	Apply ReIn.
>
> Because our work focuses on highly constrained scenarios, where modifications to model parameters and prompts are restricted, practitioners may employ ReIn as a temporary or dynamic mechanism to mitigate negative user experiences prior to implementing major updates to model parameters or prompts.

---

> > ### Comment · Reviewer_BUSn · 2025-11-27
> >
> > Thank you for your detailed and thoughtful reply. I have carefully read your reply and appreciate your patience in answering my questions.
> >
> > Resolved Issues:
> >
> > - The main performance metric is not LLM scoring. This resolves my concerns.
> >
> > - The new data on inference latency/cost in the rebuttal and the proposed parallel design address my concerns about the practical cost of deploying REIN in a production environment.
> >
> > Remaining Issues:
> >
> > - The lack of any human evaluation (even a small-scale study) remains a gap for a work whose motivation is to improve user experience.
> >
> > - The evaluation is limited to a taxonomy of 6 error types and 2 domains, and it remains unclear how well it could perform / how much more efforts are needed to apply ReIn in a production system with more diverse errors and domains. It would be better if you can demonstrate it with an example.
> >
> > The rebuttal has strengthened the paper by clarifying the evaluation method and demonstrating efficiency. I believe the paper is a solid contribution to the field, but due to the remaining issues, I will keep my rating of 6.

---

### Author Response · Authors · 2025-11-20
**Authors' General Response [1/N]**

Dear reviewers and chairs,

Thank you for providing valuable feedback to our work. As we observe a few overlapping comments among reviewers, we here write a general response for them. Our additional experiments and justifications will be reflected to the camera ready version of our manuscript.

---

## 0. Positive remarks from reviewers

We clustered notable strengths of our work and corresponding quotes from reviewers.

- **Important and realistic problem**
    - ```Reviewer BUSn```: *The paper identifies an practically important problem*
    - ```Reviewer QXtB```: *REIN addresses a highly practical and underexplored challenge*
    - ```Reviewer 4bnp```: *This paper provides a good initial exploration in this largely uncharted area.*
- **Clear writing**
    - ```Reviewer BUSn```: *The writing is clear and concise.*
    - ```Reviewer QXtB```: *Clarity and polish are consistently high throughout the manuscript. Figures and tables are well-integrated and meaningfully support the narrative.*

- **Methodological contribution**:
    - ```Reviewer BUSn```: *REIN's mechanism is clearly formalized with Algorithm 1 providing precise implementation details.*
    - ```Reviewer QXtB```:
        - *The core innovation is conceptually elegant and technically compatible with instruction hierarchies*
        - *The paper presents a clear, rigorous, and well-structured algorithm (Algorithm 1) to formalize REIN's operational flow.*
    - ```Reviewer 4bnp```: *it is impressive that the proposed method remains competitive against prompting-based baselines (...) This provides additional evidence of its utility and suggests greater flexibility in application.*
- **Experimental contribution**
    - ```Reviewer JYAX```:
        - *Goes beyond empirical results to explore theoretical grounding through instruction hierarchy analysis*
        - *Tests the approach across diverse models (...) Includes explicit seen/unseen error splits to verify the method generalizes beyond training examples to novel error types.*
    - ```Reviewer QXtB```:
        - *The empirical evaluation is particularly compelling.*
        - *Another strength lies in the careful analysis of instruction hierarchy*
        - *It also highlights a subtle but powerful mechanism of control within the existing capabilities of LLM agents.*
    - ```Reviewer 4bnp```: *The experiments also provide strong support for the effectiveness of the proposed method.*

---

> ### Author Response · Authors · 2025-11-20
> **Authors' General Response [2/N]**
>
> ## 1. Latency and cost
>
> We first clarify that the ReIn process is **analogous to Chain-of-Thought reasoning** in complex tasks. Consequently, additional latency is inherent to achieving targeted agent behaviors. A distinguishing characteristic of ReIn is that it enables capabilities beyond those originally specified in the task agent's design.
>
> In this section, we measure the latency and API cost introduced by incorporating ReIn into the task agent. For the fast experiment, we sampled 40 (2 domains * 4 tasks * 5 instances) from 588 dialogue scenarios.
>
> The tables below report the average latency and cost associated with a single ReIn process. As ReIn can operate in two modes (i.e., activated or not activated), we provide separate tables for each condition. Although no monetary cost is incurred when using local models (e.g., Llama 3B), we include the estimated cost based on current commercial API pricing for completeness and comparability.
>
> ---
>
> ### [Condition 1] ReIn is activated
>
> | Task\Model | Sonnet 3.7 | | Haiku 3.5 | | Llama 3B | |
> |:--:|:--:|:--:|:--:|:--:|:--:|:--:|
> | | Latency (sec) | Cost ($) | Latency (sec) | Cost($) | Latency (sec) | Cost($) |
> | Interpretation | 6.1074 ± 0.2675 | 0.0134 ± 0.0003 | 9.5133 ± 0.2355 | 0.0042 ± 0.0000 | 5.1282 ± 0.2376 | 0.0005 ± 0.0000 |
> | Anaphora | 6.9614 ± 0.7287 | 0.0138 ± 0.0003 | 9.2141 ± 0.7702 | 0.0041 ± 0.0001 | 4.7240 ± 0.4961 | 0.00005 ± 0.0000 |
> | Action | 6.4257 ± 0.5578 | 0.0137 ± 0.0002 | 9.7054 ± 0.4300 | 0.0043 ± 0.0001 | 5.7631 ± 0.6014 | 0.0006 ± 0.0000 |
> | Params | 6.8037 ± 0.7796 | 0.0138 ± 0.0003 | 9.5619 ± 0.6896 | 0.0043 ± 0.0001 | 4.2898 ± 0.5539 | 0.0006 ± 0.0000 |
>
> The results indicate clear differences in latency across models, although the overall pattern is largely expected. Notably, Haiku 3.5 exhibits substantially higher latency than Sonnet 3.7, despite being a smaller and less capable model. In contrast, the local Llama 3B model consistently achieves the lowest latency across all tasks. The cost estimates show minimal variation, which is unsurprising given that each ReIn operation produces only a single inception block conditioned on the input context.
>
> ---
>
> ### [Condition 2] ReIn is not activated
>
> | Sonnet 3.7 | Haiku 3.5 | Llama 3B |
> |:--:|:--:|:--:|
> | Latency (sec) | Latency (sec) | Latency (sec) |
> | 3.3637 ± 0.1456 | 4.0426 ± 0.1844 | 4.2289 ± 0.3625 |
>
> For this result, we sampled the same amount of turns where ReIn is not activated (i.e., the inception module does not identify potential issues in user utterance). It demonstrates that **the latency significantly drops** when the inception block is not generated. Given that ReIn is triggered only on turns where potential issues are expected (as shown in ```Appendix L```), the **additional latency by ReIn demonstrates a justified form of *positive friction*,** consistent with the argument presented by Inan et al. [1].
>
> ---
>
> ### Realistic low-latency system design
>
> In real-world system design, **latency can be minimized even further by executing the task agent response generation and the ReIn inception module in parallel**. Under this architecture, the system pre-computes the standard response while simultaneously evaluating whether inception is necessary. If no intervention is required, the system returns the pre-generated base response with negligible overhead. Conversely, if ReIn is activated (i.e., user-side errors are identified), the system selectively injects the inception block into their internal pipeline, thereby ensuring safety and reliability while maintaining tight latency bounds. This parallel execution strategy enables dynamic reasoning augmentation with minimal additional latency (i.e., **~4sec in both [Condition 1] and [Condition 2] can be reduced in parallel design: injecting inception block within ~2sec using sonnet 3.7**).

---

> ### Author Response · Authors · 2025-11-20
> **Authors' General Response [3/N]**
>
> ## 3. Justification for predefined taxonomy
>
> Some reviewers expressed concern that our study relies on a predefined set of error scenarios. As detailed in ```Section 3.2.1```, the six scenarios we employ are derived from a broad range of error situations *extensively examined in prior research*. **While it is not feasible to simulate every conceivable error case, constructing a representative taxonomy enables us to design controlled and systematic experiments**.
>
> We would also like to highlight the ```Reviewer 4bnp```'s comment (*"This paper provides a good initial exploration in this largely uncharted area."*). Given that our work investigates a relatively underexplored domain, we believe it offers a valuable starting point for discussing practical error recovery under realistic constraints.
>
> ---
>
>
> ## 4. Justification for human evaluation
>
> We did not adopt human evaluation in our work for the following reasons:
>
> ---
>
> ### 4.1. Objective evaluation: Verifiable evaluation signal
>
> To reliably measure the error recovery performance, we established two recovery plans: (1) an internal reporting plan for ambiguous scenarios and (2) human escalation plan for unsupported scenarios. The first plan can be straightforwardly evaluated by verifying whether the task agent generates the internal report. Similarly, the second plan can be evaluated by determining whether the task agent appropriately discontinues the service. Consequently, **the objective verification of recovery quality can be conducted** using widely adopted Pass@1 metric [2,3,4], which evaluates whether the agent successfully achieves the specified goal within a single trial (```Section 4.1```).
>
> It should be noted that recovery plans are *customizable*, as we mentioned in ```Section 3.2.2```; in this work, we designed them for **objective and consistent evaluation of the recovery quality**.
>
> ---
>
> ### 4.2. Subjective evaluation: The role of ReIn and task agent
>
> ReIn *nudges* the task agent toward a desired behavioral direction by injecting a single inception block, rather than enforcing explicit control. After this point, **all remaining behaviors are carried out entirely by the task agent**, using the same fixed parameters and system prompts as usual. Consequently, the task agent behaves identically to the original setup in every respect except for the recovery plans, **meaning both setup has similar degree of conversational quality and naturalness**.
>
> For instance, in ambiguous scenarios, both ReIn-activated and deactivated task agents will request clarification from the user; however, only ReIn-activated agent additionally produces an internal report. In unsupported scenarios, only ReIn-activated agent will escalate to a human assistant. This alloss for clear and automatable comparisons without manual human evaluation.
>
> In short, our definition and use of recovery plans enable a controlled, objective experimental setting in which **recovery performance can be evaluated without relying on human judgements**. For applications of ReIn in more realistic, non-simulated environments, where user errors arise organically rather than through explicit scenarios, we supplement our evaluation with qualitative case studies provided in ```Analyzing false positive of ReIn: without templated interaction``` and ```Appendix L```.
>
> ---
>
> ## References
>
> - [1] Inan et al. (2025) [Better Slow than Sorry: Introducing Positive Friction for Reliable Dialogue Systems](https://arxiv.org/abs/2501.17348).
> - [2] Yao et al. (2025) [$\tau$-bench: A Benchmark for Tool-Agent-User Interaction in Real-World Domains](https://openreview.net/forum?id=roNSXZpUDN)
> - [3] Chen et al. (2021) [Evaluating Large Language Models Trained on Code](https://arxiv.org/abs/2107.03374)
> - [4] Wen et al. (2025) [Program Synthesis Benchmark for Visual Programming in XLogoOnline Environment](https://aclanthology.org/2025.acl-long.769/)

---

> ### Author Response · Authors · 2025-11-20
> **Authors' General Response [2.5/N] - Sorry we omitted this!**
>
> ## 2. Analyzing false positive of ReIn: without templated interaction
>
> To address concerns regarding potential false positives in ReIn application, situations where the inception block may be injected during normal user interactions, we conduct an evaluation under realistic conditions. Specifically, we initiate conversations *without the 3-turn initial context* used for error simulation, allowing dialogues to unfold naturally from the outset. We then provide qualitative analysis of representative cases to examine the task agent's behavior in such scenarios. Full dialogue rollouts will be included in our final version.
>
> ### 2.1. Pattern 1: Not intended, but naturally occurring erroneous situations
>
> > - **User**: (...) I understand. In that case, could we wait for the refund to be processed first before making the new booking? That way I can minimize the charge on my master card.
> > - **Agent (Inception Block)**: This is an ambiguous situation where the user is asking to wait for a refund to be processed before making a new booking, but I've already completed the booking in my previous response. The user seems to have misunderstood the sequence of events that have already occurred. I need to clarify what has happened and explain why we can't "wait" for something when the action has already been completed. This is a case where the user's request contains an ambiguous understanding of the current state of their reservation.
> > - **Agent (Internal Report)**: The user has requested to wait for their refund to be processed before making a new booking, but I've already completed the booking process in my previous responses. The new reservation (HATHAT) has already been confirmed and the payment has already been processed with the higher MasterCard charge. This creates ambiguity because the user seems to be under the impression that we can still choose to delay the booking, when in fact the transaction has already been completed. The user's request indicates they didn't realize the booking was already finalized when I provided the confirmation details.
> > - **Agent**: I apologize for the confusion, but I've already completed the booking for your new business class reservation (confirmation code HATHAT). The payment has already been processed, (...)
>
> As illustrated in this example, although we did not explicitly simulate error situations, turn-taking interactions naturally give rise to potentially erroneous situations. In this specific case, the agent generates a report regarding an ambiguous situation stemming from a discrepancy in understanding between the agent and the user. An internal report generated from this situation can be utilized to design sophisticated agent behaviors by collecting diverse ambiguous situations.
>
> Similar observations can also be found in ```Section 4.5```, where ReIn dynamically decides to execute recovery plans based on naturally occuring user's unsupported requests.

---

> ### Author Response · Authors · 2025-11-20
> **Authors' General Response [2.8/N] - Sorry we omitted this!**
>
> ## 2. Analyzing false positive of ReIn: without templated interaction (Cont.)
>
> ### 2.2. Pattern 2: ReIn activation at the initial turn
>
> > - **User**: Hi there! I'd like to book the same flights as my current reservation for my friend Ivan. My user id is ivan_muller_7015.
> > - **Agent (Inception Block)**: The user's request contains ambiguity that needs to be addressed. The user wants to book "the same flights as my current reservation" for their friend Ivan, but there are multiple interpretations possible: 1. The user might be Ivan Muller (user ID ivan_muller_7015) trying to book a duplicate reservation for themselves, 2. The user might be someone else using Ivan's user ID to book a flight for a different friend also named Ivan, 3. The user might be confused about whose user ID to provide - their own or their friend's. Additionally, even if we determine whose reservation to copy, we don't have a specific reservation ID to reference. The user has only provided a user ID, but they might have multiple reservations in the system. This ambiguity needs to be clarified before proceeding with the booking process. I should ask specific questions to determine - Whose user ID has been provided - Which specific reservation they want to duplicate - Confirmation of passenger details for the new booking. This situation should be reported internally to help improve the system's handling of ambiguous booking requests.
> > - **Agent (Internal Report)**: The user's request contains ambiguity regarding booking "the same flights as my current reservation" for a friend named Ivan while providing user ID ivan_muller_7015. It's unclear if: 1. The user is Ivan Muller trying to book a duplicate reservation for themselves 2. The user is someone else using Ivan's account to book for another person named Ivan 3. The user is confused about whose user ID to provide. Additionally, no specific reservation ID was provided to reference which flights to duplicate, and the user might have multiple reservations in the system.
>
> In certain initial turns, the inception module activates ReIn due to the inherent incompleteness of initial user queries, which frequently omit necessary details. While the generated internal report may provide valuable observational insights regarding user request patterns at dialogue initiation, we acknowledge that this activation may introduce additional response latency during the first turn of interaction.
>
> An important takeaway is that **false positives do not necessarily constitute detrimental outcomes**. Given that the primary objective of recovery plans is to mitigate negative user experiences, we observe minimal instances where ReIn activation adversely affects conversational flow. Furthermore, we posit that customization of the inception module (e.g., through fine-tuning or prompt engineering) can effectively regulate the frequency of ReIn activation.

---

### Meta-Review · Area_Chair_hmie · 2026-01-08

**Summary:**

This paper presents an initial exploration of a practical yet underexplored problem: error recovery in multi-round user-agent interactions, particularly in scenarios where the agent must handle ambiguous or unsupported user requests. Reviewers generally acknowledge the novelty of the problem and the reasonableness of the proposed approach.

It appears that concerns regarding latency and cost, potential false positives, the predefined taxonomy, and human evaluation were adequately addressed in the author rebuttal. While some concerns about the sufficiency of the evaluation remain, I believe that the novelty of the problem and the proposed approach outweigh this shortcoming.

**Reviewer Concerns:**

The authors have satisfactorily addressed concerns related to latency and cost, potential false positives, the predefined taxonomy and human evaluation. However, the issue of insufficient evaluation remains only partially resolved. This limitation is understandable, given the novelty of the research problem and the relatively underexplored nature of the domain.

**Reviewer Scores:**

This paper initially received two positive ratings of 6. After the rebuttal and subsequent clarifications, it appears likely that these positive ratings will be maintained. Additionally, Reviewer 4bnp may consider increasing their rating from 4 to 6. As a result, the paper may ultimately receive three positive ratings.

---

### Decision · Program_Chairs · 2026-01-26

Accept (Poster)